

# Advancing CH$_4$ and N$_2$O retrieval strategies for NDACC/IRWG high-resolution direct-sun FTIR observations

Ivan Ortega[1], James W. Hannigan[1], Bianca C. Baier[2], Kathryn McKain[2], and Dan Smale[3]

[1]Atmospheric Chemistry Observations & Modeling, NSF National Center for Atmospheric Research, Boulder, Colorado, USA
[2]Global Monitoring Laboratory, National Oceanic and Atmospheric Administration, Boulder, Colorado, USA
[3]National Institute of Water and Atmospheric Research (NIWA), Lauder, New Zealand

**Correspondence:** Ivan Ortega (iortega@ucar.edu)

**Abstract.**

Atmospheric methane (CH$_4$) and nitrous oxide (N$_2$O) are potent greenhouse gases with significant impacts on climate change. Accurate measurement of their atmospheric abundance is essential for understanding their sources, sinks, and the impact of human activities on the atmosphere. Ground-based high-resolution Fourier Transform Infrared (FTIR) observations, employed by collaborative international initiatives like the Infrared Working Group (IRWG) within the Network for the Detection of Atmospheric Composition Change (NDACC), play a vital role in retrieving the atmospheric amounts of these gases. Network wide consistent data products rely on consistent observations and retrievals. Recent developments in spectroscopy, a priori data, retrieval software and techniques underscores the necessity to revisit the retrieval strategies for all NDACC/IRWG species currently ongoing. This study investigates various retrieval strategies of CH$_4$ and N$_2$O utilizing high-resolution FTIR observations in Boulder, Colorado, and compares them with unique airborne in situ measurements. The initial focus is on characterizing retrieval differences across spectroscopy databases. While it is challenging to identify the best retrievals purely based on spectroscopy, as they produce similar outcomes, notable differences in profile shapes and magnitudes underscore the importance of independent validation. Specifically, when multi-year independent nearby AirCore and aircraft in situ profile measurements are used to evaluate vertical distributions and biases in partial columns, they reveal excellent agreement in relative differences with FTIR retrievals and thereby strengthening confidence in the assessment. The final optimized retrievals for CH$_4$ and N$_2$O are presented incorporating quantitative fitting results and comparisons of vertical profiles, partial and total columns. We find that employing a priori profiles using the latest simulations of the Whole Atmosphere Community Climate Model (WACCM) enhances accuracy relative to in situ profiles. While the HITRAN 2020 spectroscopic database is effective for N$_2$O, ATM 2020 provides better results for CH$_4$, with slight improvement observed when paired with the water vapor line list from DLR; however, this improvement may be site-dependent. Regarding regularization, both first-order Tikhonov and Optimal Estimation produce comparable outcomes, as long as the fitted profile degrees of freedom remain between 2 and 2.5. Correspondingly, profile results comparisons yield biases of -0.08 $\pm$ 0.38 % and 0.89 $\pm$ 0.28 % for tropospheric and stratospheric layers of CH$_4$ relative to AirCore, respectively, and 0.39 $\pm$ 0.42 % for aircraft comparisons in the troposphere. For N$_2$O, the bias in the troposphere using aircraft measurements is approximately 0.18 $\pm$ 0.2 %. Uncertainty budgets combining random and systematic sources are provided. Random errors, mainly stemming from temperature profile uncertainties and mea-





surement noise dominate in the troposphere for both gases with a retrieval random error of 0.5 %. Systematic errors primarily arise from HITRAN based spectral line parameters, predominantly the line intensity and air-broadened half-width. Finally, we present long-term time series of $CH_4$ derived from the recommended retrieval strategies applied to observations at Boulder. To contrast these findings with the southern hemisphere, we successfully extended this analysis to the site in Lauder, New Zealand.

These findings contribute to advancing our understanding of atmospheric composition and will support the improvement of a harmonized approach for all IRWG/NDACC sites.

## 1 Introduction

Atmospheric methane ($CH_4$) and nitrous oxide ($N_2O$) are potent greenhouse gases that significantly contribute to radiative forcing and, consequently, climate change. Accurate and precise measurement of these trace gases is essential for understanding

their sources, sinks, and the impact of anthropogenic activities on the Earth's atmosphere. $CH_4$ is considered a short-lived greenhouse gas with an atmospheric lifetime of approximately 10 years. Despite its short atmospheric lifetime compared to other greenhouse gases, $CH_4$ has a high global warming potential, making it a significant contributor to near-term climate forcing. Its sources are diverse, including natural processes such as wetlands and anthropogenic activities such as livestock management and fossil fuel extraction (Canadell et al., 2021). In contrast, $N_2O$ is classified as a long-lived greenhouse gas,

with a lifetime of approximately 114 years. While less abundant, $N_2O$ exhibits a remarkable warming potential, approximately 270 times greater per ton than $CO_2$ over a 100-year time horizon. This amplifies its impact on climate change and its role in stratospheric ozone depletion (Prather et al., 2015; Tian et al., 2020). Understanding the sources and sinks of both $CH_4$ and $N_2O$ is crucial for developing effective climate mitigation strategies and policies. Moreover, as both gases are intricately linked to the global carbon and nitrogen cycles, studying their behavior aids in comprehending broader ecological processes

and human-induced perturbations to the Earth's atmosphere (Canadell et al., 2021).

Ground-based Fourier Transform Infrared (FTIR) spectroscopy has proven to be a reliable method for extracting atmospheric concentrations of both $CH_4$ and $N_2O$ through direct-sun solar absorption measurements. Collaborative international initiatives actively use FTIR observations to retrieve information on these species. The Infrared Working Group (IRWG) within the Network for the Detection of Atmospheric Composition Change (NDACC) has demonstrated the capability to retrieve both

gases, along with numerous others, in the mid-infrared with higher spectral resolution and for long-term (De Mazière et al., 2018). Simultaneously, the Total Column Observation Network (TCCON) focuses on precision measurements of column-averaged abundances for $CH_4$ and $N_2O$ in the near-infrared spectral region (Wunch et al., 2011). Additionally, the newly formed Collaborative Carbon Column Observing Network (COCCON) has employed portable low-resolution FTIR spectrometers to measure greenhouse gases, including $CH_4$ and $N_2O$ (Hase et al., 2016; Alberti et al., 2022). While TCCON and COCCON

have optimized their retrievals through profile scaling to derive total columns, the IRWG/NDACC is able to obtain vertical profile information with higher spectral resolution, up to two or three degrees of freedom (DOF).

Presently, sites within IRWG/NDACC generally adhere to the recommendations outlined in Sussmann et al. (2011). However, as evidenced by Sepúlveda et al. (2014); Bader et al. (2017); and Olsen et al. (2017), a significant variability exists in



the final retrieval strategies employed at each site. The retrieval strategy proposed by Sussmann et al. (2011) for CH$_4$ involved using the HIgh-resolution TRANsmission molecular absorption database (HITRAN) 2001, which exhibited better agreement between measured and simulated spectra compared to newer databases of that time (e.g., HITRAN 2008, (Rothman et al., 2009)). This strategy utilized three out of five possible micro windows and incorporated Tikhonov L1 regularization, which focuses on stabilizing the solution by adding constraints to the optimization problem. In a more recent study, Chesnokova et al. (2020) analyzed total columns derived from various line lists, concluding that HITRAN 2001 offers better CH$_4$ retrievals. However, ATM19 (Toon, 2015; Toon et al., 2016) produced slightly different, yet comparable, results to HITRAN 2001. This conclusion is based on the optimal agreement observed between measured and simulated spectra. Considering the availability of updated spectroscopy, diverse retrieval strategies at each site, and new validation data, there is a pressing need to explore an optimal and harmonized strategy within the IRWG/NDACC. Additionally, while previous studies have focused mainly on total columns, IRWG/NDACC observations have the potential to separate both tropospheric and stratospheric columns for N$_2$O and CH$_4$. However, this has not yet been extensively validated.

This study investigates diverse retrieval strategies for CH$_4$ and N$_2$O, employing several micro-windows, a priori profiles, and regularization methods. High-resolution Fourier-transform infrared (HR-FTIR) observations in Boulder, Colorado form the basis of our analysis (Ortega et al., 2021). Initially, we focus on characterizing residuals, representing differences between simulated and observed spectra using different spectroscopy databases. A key aspect of our investigation involves integrating multi-year independent observations from AirCore (CH$_4$) and aircraft (CH$_4$ and N$_2$O) profiles obtained in close proximity to the FTIR site. These independent datasets are crucial for evaluating the vertical distribution (profile shapes) of the retrieval strategies. For CH$_4$, we examine the bias in tropospheric and stratospheric partial columns, while for N$_2$O, the assessment primarily focuses on tropospheric columns. Boulder offers a unique opportunity with its extensive, multi-year coverage of semi-co-located AirCore and IRWG/NDACC profiles, allowing for the integration of numerous observations into our analysis.

## 2  Observations

### 2.1  Ground-based FTIR

High-resolution (HR) direct-solar infrared absorption spectra have been recorded at the NSF National Center for Atmospheric Research (NSF NCAR) in Boulder, Colorado (40.40° N, 105.24° W, 1600 m.a.s.l) since 2010 to present, from 2010 to 2017 utilizing a Bruker 120 HR-FTIR spectrometer. Subsequently, the instrument underwent an upgrade throughout a significant portion of 2018, incorporating the latest electronics and optics from the Bruker 125 HR-FTIR. Measurements are conducted under clear-sky conditions. The site officially became a part of the Infrared Working Group (IRWG) within NDACC in 2020, and the observations adhere to the standard measurement protocols established by the network. Optical bandpass filters are employed to optimize the signal-to-noise ratio (SNR) within the near and mid-infrared spectral range. For CH$_4$ NDACC filter number 3 is used while for and N$_2$O filters 3 and 4 can be used. The nominal spectral resolution achieved is 0.004 cm$^{-1}$ (optical path difference (OPD) of $\approx$ 257 cm), utilizing liquid nitrogen-cooled InSb and Mercury Cadmium Telluride (MCT) detectors and a KBr beam splitter. For a detailed description of the FTIR operations at Boulder see Ortega et al. (2019, 2021).



The spectra undergo analysis using the SFIT4 algorithm, an advancement from SFIT2 (Pougatchev et al., 1995; Rinsland et al., 1998; Hase et al., 2004; Hannigan et al., 2024), to extract profiles of $CH_4$ and $N_2O$. Key input and common parameters for the retrieval of all gases using SFIT4 include vertical profiles of pressure, temperature, and volume mixing ratios (VMR)

of the atmospheric gases absorbing in the specific micro-windows. The input pressure and temperature vertical profiles are obtained from National Center for Environmental Prediction (NCEP) (Wild et al., 1995) and are standardized for use in all NDACC retrievals. These are daily average profiles that extend to up approximately 50 km and above that we use monthly mean pressure and temperature profiles from simulations using the Whole Atmosphere Community Climate Model (WACCM) (Garcia et al., 2007). As an initial step, and to mitigate water vapor interference, the retrieval of water vapor is conducted first.

Subsequently, this water vapor information is utilized in the retrieval process for $CH_4$ and $N_2O$ to prevent potential bias arising from inaccuracies in the water vapor profile shape (Ortega et al., 2019). Sections 3.1 and 3.2 describe in more detail the specific retrievals for $CH_4$ and $N_2O$, respectively.

## 2.2 AirCore and aircraft profiles

We employ $CH_4$ AirCore data acquired from the National Oceanic and Atmospheric Administration (NOAA) Global Moni-

toring Laboratory (GML) to examine and assess the vertical distribution of $CH_4$ in the troposphere and stratosphere. Regular monthly AirCore flights are initiated in Boulder, Colorado, many of these flights occur at approximately 1 pm local time to synchronize with the OCO-2/A-train satellite overpass with air samples collected anywhere from 17 km to 121 km nearby to the FTIR (measured from the descent point). The AirCore technique employed by NOAA involves collecting air samples within a stainless steel tube (approx. 100 m in length and 1 cm in diameter). This tube passively collects atmospheric air

during balloon descents from the middle stratosphere to the ground. Subsequently, the collected air is analyzed using a cavity ring-down spectrometer ensuring a trace mole fraction precision of better than 0.4 ppb for $CH_4$ (Karion et al., 2010). Because storage of the air sample in the tubing does not diffuse quickly between balloon payload landing and analysis, approximately 100 discrete samples are measured in the AirCore tubing in a 4-hour storage time, offering detailed insights into the vertical distribution of $CH_4$, carbon monoxide (CO), and carbon dioxide ($CO_2$). For our study, we leverage Level 2 $CH_4$ from the

current NOAA AirCore dataset v20230831, which includes dry air mole fraction profiles retrieved since 2012 (Baier et al., 2021). Our investigation encompasses profiles collected from 2018 to 2022, allowing us to evaluate various retrieval strategies with a substantial number of observations, thereby facilitating the creation of statistically meaningful findings.

Figure 1a shows individual $CH_4$ AirCore profiles obtained in Boulder, Colorado from 2018 to 2022. As expected, there is a noticeable rise in $CH_4$ mole fraction from 2018 onwards, reflecting the increase in the global background (https://gml.

noaa.gov/ccgg/trends_ch4/). These profiles exhibit distinct features, such as elevated near-ground values in certain instances, suggestive of local sources. Further, within the boundary layer and free troposphere, the profiles demonstrate well-mixed characteristics with some variations in the upper troposphere and the lowest stratosphere of the AirCore profiles, culminating in a sharp decrease at the tropopause. Figure 1b presents the mean vertical profiles from 2018 to 2022. To derive the mean profile, the individual profiles are re-gridded to a common altitude levels within a range from 1.6 to 25 km and mean values

are then estimated within 0.2 km resolution. Ultimately, mean and standard deviation profiles are calculated. The blue shaded





area in Figure 1b represents the standard deviation in the AirCore profile dataset used here, while the red circle at the bottom represents the mean values from an in-situ sensor located at the Boulder reservoir about 5 km from the HR-FTIR using the overlap period between 2020 to 2022 (https://www.bouldair.com/boulder.htm), providing insight into the variability around the Boulder region.

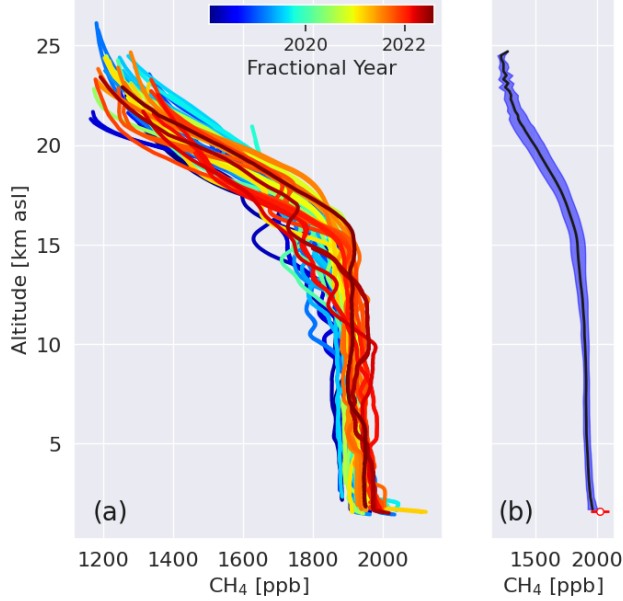

**Figure 1.** (a) Individual AirCore $CH_4$ profiles in Boulder, Colorado, color-coded to indicate fractional years spanning from 2018 to 2022. (b) Mean vertical profile of $CH_4$ accompanied by standard deviation, with the red circle at the bottom representing the mean volume mixing ratio from an in-situ sensor near the HR-FTIR (see text for details). A total of 61 vertical profiles are included in these plots.

To further evaluate $CH_4$ and $N_2O$ retrievals we use the vertical profiles obtained in a remote location nearby Briggsdale, Colorado (40.6347 N, 104.3269 W), approximately 100 km northeast of the HR-FTIR, as part of the NOAA/ESRL Global Greenhouse Gas Reference Network Aircraft Program (Sweeney et al., 2015; McKain et al., 2023) (https://gml.noaa.gov/ccgg/ aircraft/). This program employs contracted aircraft equipped with equipment to collect 12 discrete air samples in vertical profiles, primarily over North America, and monitor over 50 trace gases throughout the boundary layer and free troposphere over

the long term. The network provides crucial insights into the sources, sinks, and transport mechanisms of greenhouse and other trace gases. Additionally, these profiles are a resource for model evaluation of vertical mixing and independently constrain the mean biospheric uptake of $CO_2$ throughout the annual cycle. The routine data collection within the program further facilitates the evaluation of satellite retrievals, which is often complemented by ground-based remote sensing observations. Figure 2 illustrates the $CH_4$ and $N_2O$ aircraft profiles acquired at Briggsdale, Colorado, spanning from 2018 to 2022. In line with

$CH_4$ AirCore profiles, there is minimal variation observed in the profiles up to 6 km, with occasional enhancements near the ground on certain days. In the case of $N_2O$, there is a clear increase from 2018 onwards, also consistent with the increase in the





global background (https://gml.noaa.gov/ccgg/trends_n2o/). The N$_2$O profiles exhibit a well-mixed pattern, with no notable enhancements near the ground. Similar to Figure 1, the mean profiles are presented on the right side for each gas, depicting mainly the increase in the global background. However, the mean profile is displayed within the altitude range of 1.6 to 7 km asl, with a resolution of 0.4 km due to the original vertical resolution being lower than that of the AirCore.

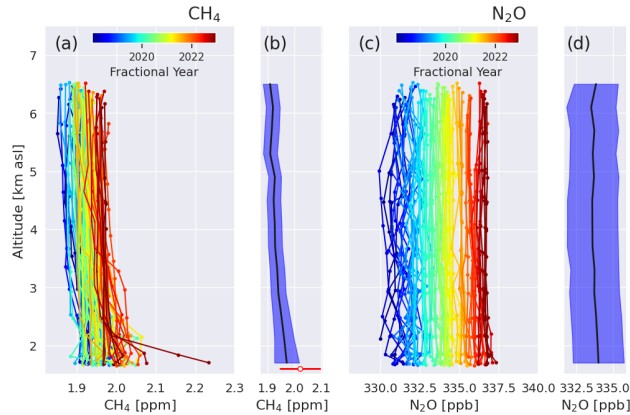

**Figure 2.** (a) NOAA aircraft profiles of CH$_4$ at Briggsdale, Colorado, with different colors indicating fractional years between 2018 and 2022. (b) Mean and standard deviation of CH$_4$ profiles derived from the NOAA aircraft, along with surface CH$_4$ volume mixing ratio (vmr) from the Boulder reservoir, including standard deviation. (c) Same as (a) but for N$_2$O. (d) Same as (b) but for N$_2$O. A total of 81 vertical profiles are included in these plots.

## 3 Retrieval strategies

### 3.1 CH$_4$

Multiple retrieval strategies have been devised, encompassing various combinations of micro-windows, spectroscopy, and regularization techniques. The fitting results are employed to assess different spectroscopy databases, and all retrieval strategies are compared with AirCore and aircraft observations to determine the most effective approach. An optimized selection of micro-windows (MW) in the mid-infrared has been carried out by Sussmann et al. (2011) and a summary of these retrieval strategies is provided in Table 2. In alignment with Sussmann et al. (2011), we employed the same five potential micro-windows for our analysis, detailed in Table 1.

The molecular absorption databases employed in this study include the latest official release of HITRAN 2020 (HIT20) (Gordon et al., 2022), the ATM 2020 (ATM20) spectroscopic database used by TCCON (Toon, 2015; Toon et al., 2016), which is available at https://mark4sun.jpl.nasa.gov/toon/linelist/linelist.html (last accessed: 22 Dec 2023) (last accessed: 22 Dec 2023), and HITRAN 2000 (HIT00) (including the 2001 updates), as recommended by Sussmann et al. (2011). In their study, Sussmann et al. (2011) also tested HITRAN 2008, which was the most recent database at the time, and found significant





residuals with HITRAN 2008, while HIT00 showed much lower residuals. More recently, Chesnokova et al. (2020) examined
the impact of different databases and similarly concluded that HIT00 provides superior fitting results, with the exception
of ATM, 2019 at the time. Additionally, we utilized combinations of these databases with others for specific purposes. For
instance, a water vapor spectroscopy is based on the line list provided by the German Aerospace Center (DLR) (Zhou et al.,
2018), while ATM20 is used in conjunction with the HIT00 linelist for $CH_4$. Section 4.1 provides a more detailed overview of
the various spectroscopy databases and their corresponding results.

For the current IRWG gas retrievals, mean a priori profiles were primarily obtained using WACCM version 4 for the 1980
- 2020 period (referred to as WACCM4). WACCM simulations underwent enhancements in physical, chemical, and aerosol
parameterizations (Gettelman et al., 2019), resulting in version 6 (WACCM6) and new mean a priori profiles have been created
spanning the 1980 - 2040 period (https://wiki.ucar.edu/pages/viewpage.action?pageId=452264118). Comparing $CH_4$ between
these two WACCM versions at different sites (e.g., Mauna Loa, Hawaii; Thule, Greenland) with independent profiles obtained
during the airborne campaign Pole-to-Pole Observations (HIPPO; www.eol.ucar.edu/field_projects/hippo) (Wofsy, 2011), and
Atmospheric Tomography Mission (ATom; https://daac.ornl.gov/ATOM/campaign/) (Thompson et al., 2022) field campaigns
indicates that the latest version 6 profiles are more suitable for recent years (not shown). To streamline the retrieval process
all retrievals utilized the most recent version of WACCM profiles. For regularization, the process of adding constraints or
penalties to the optimization problem to ensure a more stable and reliable solution, we employed both the first-order Tikhonov
regularization, following the recommendation of Sussmann et al. (2011), and the conventional optimal estimation (OEM)
approach. In Tikhonov regularization, the regularization strength ($\alpha$) is optimized to maintain degrees of freedom (DOF) at
approximately 2, with potential variations depending on conditions. Optimization is achieved through the typical L-curve
method, as proposed by Steck (2002). The Tikhonov regularization procedure is directly implemented in the latest version of
SFIT4, accessible at https://wiki.ucar.edu/display/sfit4/SFIT+Core+Code (last accessed: 22 Dec 2023). For the OEM approach,
regularization involved diagonal a priori covariance matrices. Variances magnitudes were empirically tuned to prevent profile
oscillations and similarly to achieve a target DOF of around 2. The EOM technique has the advantage that the retrieved vertical
profile relaxes back to the a priori profile shape, as spectral information diminishes with altitude. In contrast, the Tikhonov
constraint tends to preserve the deviation from the a priori at higher altitudes where spectral information is still present.
Additional details on the retrieval workflow in Boulder are provided in Ortega et al. (2021).

We establish a baseline retrieval strategy (r1) by employing three micro-windows (1, 3, and 5), along with the first-order
Tikhonov regularization approach, and the HIT00 spectroscopy database (Sussmann et al., 2011). A total of ten retrieval
strategies have been developed to explore various approaches and determine whether different methods impact the results
when compared with AirCore and aircraft profiles. As shown in Section 4.1, HIT20 was excluded due to its significantly
higher residuals in the fitting results. Detailed information on each retrieval strategy is provided in Table 2.

## 190    3.2    $N_2O$

The $N_2O$ retrievals suggested by the IRWG involve the use of four micro-windows within the 2480 – 2540 $cm^{-1}$ spectral
range. The specific micro-windows and associated interfering species are detailed in Table 3. Our focus is not directed towards





**Table 1.** Summary of the 5 micro windows tested in the the retrieval of $CH_4$.

| | Micro-windows ($cm^{-1}$) | Main species |
|---|---|---|
| 1 | 2613.70 – 2615.40 | $CH_4$, HDO, $CO_2$ |
| 2 | 2650.60 – 2651.30 | $CH_4$, HDO, $CO_2$ |
| 3 | 2835.50 – 2835.80 | $CH_4$, HDO |
| 4 | 2903.60 – 2904.03 | $CH_4$, $NO_2$, HDO, $H_2O$ |
| 5 | 2921.00 – 2921.60 | $CH_4$, $H_2O$, HDO, $NO_2$ |

**Table 2.** Different strategies tested in the retrieval of $CH_4$.

| Retrieval | MWs | Spectroscopy details | Regularization | Additional details |
|---|---|---|---|---|
| r1 | 1,3,5 | HIT00 | Tikhonov L1 | Baseline following Sussmann et al. (2011) |
| r2 | 1,3,5 | ATM20 | Tikhonov L1 | ATM20 for all gases |
| r3 | 1,3,5 | ATM20 and HIT00 ($CH_4$) | Tikhonov L1 | Same as r2 but HIT00 for $CH_4$ |
| r4 | 1,2,3,4,5 | ATM20 and HIT00 ($CH_4$) | Tikhonov L1 | 5 Micro-windows |
| r5 | 1,2,3,4,5 | ATM20 | Tikhonov L1 | 5 Micro-windows and ATM20 for all gases |
| r6 | 1,2,3,5 | ATM20 | Tikhonov L1 | 4 Micro-windows and ATM20 for all gases |
| r7 | 1,2,3 | ATM20 | Tikhonov L1 | 3 Micro-windows and ATM20 for all gases |
| r8 | 1,2,3,4,5 | ATM20 and DLR ($H_2O$) | Tikhonov L1 | DLR water vapor and ATM20 all others |
| r9 | 1,2,3,4,5 | ATM20 and DLR ($H_2O$) | OEM | same as r8 but using OEM |
| r10 | 1,2,3,4,5 | ATM20 | Tikhonov L1 | Same as r5 but using monthly mean WACCM6 |

the identification or refinement of micro-windows since they are well-harmonized across the network, yielding satisfactory results. Instead, our objective is to assess the performance of the latest spectroscopy. Similarly, the spectroscopy evaluation includes testing HITRAN 2008 (HIT08) (the current official IRWG recommendation), HIT20, and ATM20. Within the retrieval strategies outlined in Table 4, we incorporate both the Tikhonov L1 and OEM approaches since the regularization approach is not uniformly harmonized (Zhou et al., 2019). Additionally, two optical band pass filters, denoted as filters 3 and 4 within the IRWG, are available for utilization in the $N_2O$ retrieval process. In this case, we conducted separate $N_2O$ retrievals using each filter independently, as well as employing a retrieval strategy where both filters are utilized simultaneously. In this instance, we incorporate retrieval strategies utilizing both WACCM4 and v6, as there is a comparatively less number of sensitivity studies compared to $CH_4$.





**Table 3.** Summary of the 4 micro windows used in the the retrieval of N$_2$O.

| | Micro-windows (MW) | Main species |
|---|---|---|
| 1 | 2481.30 – 2482.60 | N$_2$O, H$_2$O, CO$_2$, CH$_4$ |
| 2 | 2526.40 - 2528.20 | CH$_4$, N$_2$O, CO$_2$, HDO, CH$_4$ |
| 3 | 2537.85 - 2538.80 | N$_2$O, CH$_4$ |
| 4 | 2540.10 - 2540.70 | N$_2$O |

**Table 4.** Different strategies tested in the retrieval of N$_2$O.

| Retrieval | Spectroscopy details | Regularization | Additional details |
|---|---|---|---|
| r1$^*$ | HIT08 | OEM | filter 3 and WACCM4 |
| r2 | HIT20 | OEM | filter 3 and WACCM4 |
| r3 | ATM20 | OEM | filter 3 and WACCM4 |
| r4 | HIT20 | OEM | filter 4 and WACCM4 |
| r5 | HIT08 | OEM | filter 3 and WACCM6 |
| r6 | HIT20 | Tikhonov L1 | filter 3 and WACCM6 |
| r7 | HIT20 | Tikhonov L1 | filter 4 and WACCM6 |
| r8 | HIT20 | Tikhonov L1 | filters 3, 4 and WACCM6 |

$^*$Current retrieval strategy as suggested by the IRWG

# 4 Results

## 4.1 Inter-comparison among spectroscopy databases

This section addresses the quality of fits resulting from various spectroscopy databases. Specifically, for CH$_4$ we compare
five databases: (1) HIT00; (2) HIT20; (3) ATM20; (4) ATM20 with DLR water vapor (ATM20_DLRWV); and (5) ATM20
with HIT00 for CH$_4$ (ATM20_HIT00CH4)).To maintain focus on spectroscopy, all five retrievals follow the same procedure,
employing Tikhonov regularization and WACCM6 as the a priori, with the only variation being the choice of spectroscopy.
Figure 3 visually shows the spectral contributions of various species during the CH$_4$ retrieval across the five micro-windows.
As an example, the top panel displays these contributions for each micro-window on November 18th, 2019, at a solar zenith
angle of 60.0°. The observed and fitted lines, presented in blue and green, respectively, provide insights into the quality
of the retrieval. In the corresponding bottom plots, the residuals (observed minus fitted) are presented, with distinct colors
indicating the different spectroscopy databases. There are clear differences for some micro-windows. For instance, HIT20
exhibits substantial systematic residuals for micro-windows 2 and 5. The large residual at 2921.33 cm$^{-1}$ in HIT20 for micro-
window 5, previously identified by Sussmann et al. (2011) for HIT08, suggests that HIT20 still needs to be improved for
this line strength. To further identify differences, the numbers within the labels provide the root mean square fitting for each
micro-window across diverse spectroscopy databases. Micro-window 1 stands out as the only one where all databases perform



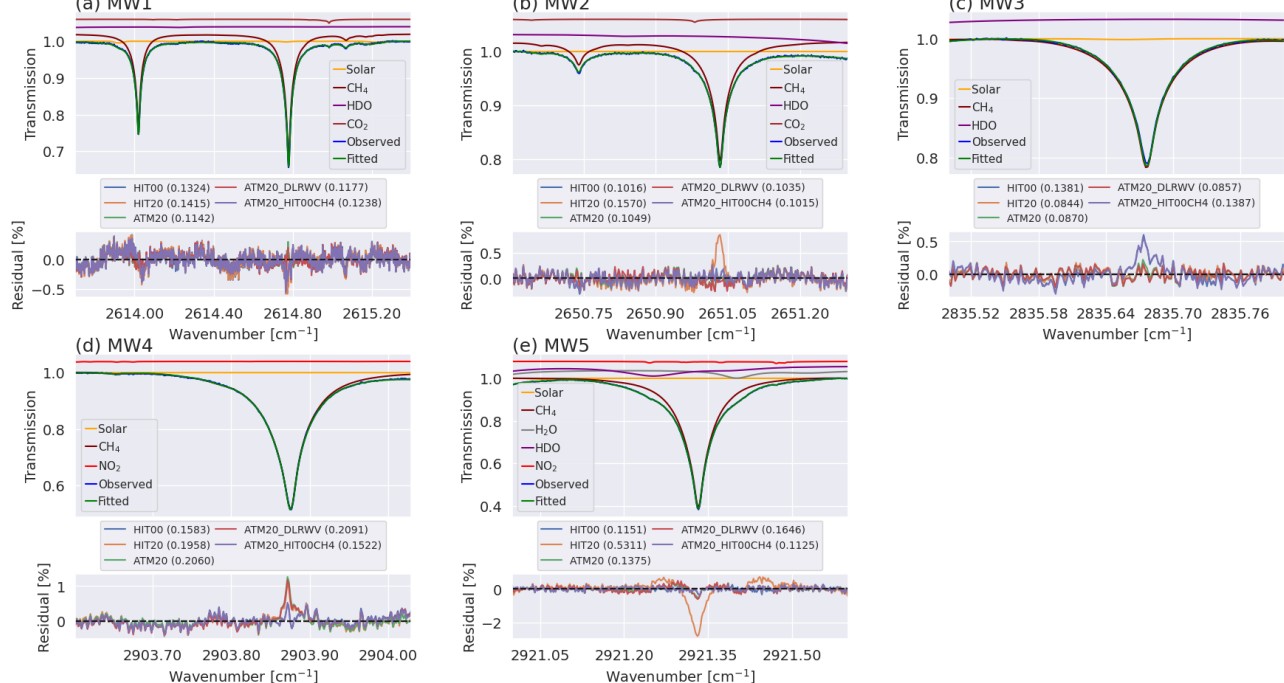

**Figure 3.** Retrieval fit example of $CH_4$ for November 18th, 2019, with a solar zenith angle of $60.0°$. Spectral contributions for various species are displayed above each micro-window, spanning from micro-window 1 (a) to micro-window 5 (e) employing Tikhonov regularization and WACCM6 as the a priori. The observed and fitted lines are represented in blue and green, respectively. The bottom plots shows the residuals (observed minus fitted), with distinct colors indicating different spectroscopy types as labeled. The numbers within the labels denote the root mean square fitting for each micro-window across various spectroscopy databases.

similarly, while others exhibit variations. Nonetheless, ATM20 demonstrates slightly better results within micro-window 1. Apart from HIT20, which displays high residuals in micro-window 2, all other databases show similar residuals. In micro-window 3, ATM20, ATM20_DLRWV, and HIT20 yield improved results, while micro-window 4 shows superior residuals

with ATM20_HIT04CH4, HIT20_HIT04CH4, and HIT00, suggesting that HIT00 remains preferable for this micro-window. As previously noted, HIT20 continues to show large residuals in micro-window 5, whereas HIT00 still provides better residuals. A summary of root mean square values (RMS) for all micro-windows for a full-year (2021) of retrievals is provided in Table 5. Overall, HIT20 displays larger RMS, suggesting it may not be an optimal choice, while all other spectroscopy databases show similar RMS values within the standard deviation. These findings align with Chesnokova et al. (2020), where HIT00 yields

better RMS but is similar to ATM20. However, there are notable differences in the profile shapes and magnitudes (see section 4.2), highlighting the importance of the assessment conducted with independent observations.

For $N_2O$, we compare the databases HIT08, HIT20, and ATM20. Our objective is to assess whether HIT20 or ATM20 exhibits better residual quality than HIT08, currently employed in the IRWG/NDACC strategy. Figure 4 illustrates the spectral



**Table 5.** Overview of the mean root mean square (RMS) values across all five micro-windows and the overall RMS for $CH_4$ retrieval using various spectroscopy databases for 2021. The last column displays the degrees of freedom (DOF). Values in parenthesis are standard deviation.

| | RMS | | | | | | |
|---|---|---|---|---|---|---|---|
| Spectroscopy | MW1 | MW2 | MW3 | MW4 | MW5 | Overall | DOF |
| HIT00 | 0.140 (0.022) | 0.125 (0.022) | 0.160 (0.038) | 0.169 (0.033) | 0.168 (0.038) | 0.156 (0.030) | 2.46 (0.15) |
| HIT20 | 0.148 (0.024) | 0.164 (0.035) | 0.115 (0.016) | 0.205 (0.042) | 0.504 (0.058) | 0.270 (0.039) | 2.31 (0.13) |
| ATM20 | 0.139 (0.018) | 0.122 (0.017) | 0.116 (0.018) | 0.204 (0.036) | 0.183 (0.032) | 0.161 (0.025) | 2.42 (0.16) |
| ATM20_DLRWV | 0.140 (0.017) | 0.134 (0.022) | 0.117 (0.018) | 0.206 (0.039) | 0.210 (0.051) | 0.170 (0.032) | 2.41 (0.16) |
| ATM20_HIT00CH4 | 0.143 (0.022) | 0.120 (0.019) | 0.160 (0.037) | 0.164 (0.031) | 0.142 (0.024) | 0.150 (0.027) | 2.47 (0.14) |

contributions of various species during the $N_2O$ retrieval across four micro-windows. Similar to $CH_4$, the top panels showcase an example, specifically on January 31, 2019, at a zenith angle of 63.8°. The bottom residual plots do not reveal a significant difference among the three databases. While the RMS suggests a slight improvement with HIT20, an overview table (6) presenting results for the entire year (2021) indicates no significant improvement. Nonetheless, this suggests that either HIT20 or ATM20 could be considered in the IRWG recommendation.

**Table 6.** Summary of the RMS values for $N_2O$ retrieval in 2021, encompassing all five micro-windows and the overall RMS. The last column presents the DOF, with standard deviation values shown in parentheses.

| | RMS | | | | | |
|---|---|---|---|---|---|---|
| Spectroscopy | MW1 | MW2 | MW3 | MW4 | Overall | DOF |
| HIT08 | 0.149 (0.031) | 0.130 (0.019) | 0.133 (0.029) | 0.132 (0.032) | 0.148 (0.030) | 2.56 (0.11) |
| HIT20 | 0.148 (0.031) | 0.131 (0.019) | 0.131 (0.028) | 0.128 (0.032) | 0.147 (0.029) | 2.57 (0.11) |
| ATM20 | 0.149 (0.031) | 0.130 (0.019) | 0.133 (0.029) | 0.131 (0.032) | 0.148 (0.029) | 2.57 (0.11) |

## 4.2    Inter-comparison with AirCore and aircraft profiles

Between 2018 and 2022, a total of 61 AirCore sampling systems were launched in Boulder, 36 which were coincident FTIR measurements. The total number of coincident dates between FTIR and aircraft profiles is 51. To enable a quantitative comparison of insitu profiles and partial columns, vertical profile measurements from AirCore and aircraft are re-gridded onto the FTIR retrieval's altitude grid using spline order 1 interpolation. Subsequently, these profiles are smoothed with the daily mean FTIR averaging kernels to account for the FTIR altitude sensitivity (Rodgers and Connor, 2003). Figure 5a presents an illustra-
tive comparison between FTIR profiles obtained on a given day and an AirCore profile. The high vertical resolution AirCore profile is depicted with black triangles, while the smoothed AirCore profile is shown with blue triangles. FTIR retrievals from the same day are represented by circles in various colors. In Figure 5b, the relative difference in percentage (or bias), defined as the difference between FTIR and AirCore values divided by the AirCore values, is shown for each corresponding time. To





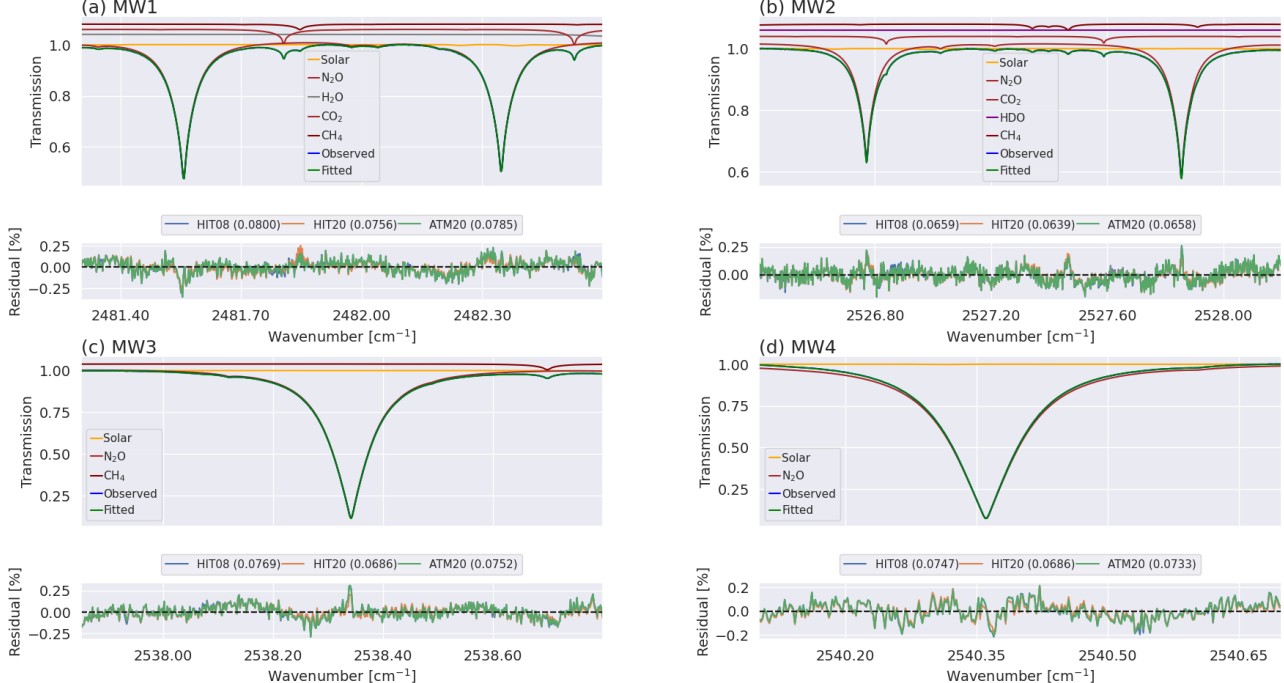

**Figure 4.** Retrieval fit example of $N_2O$ on January 31th, 2019 with a solar zenith angle of $63.8°$. Spectral contributions for various species are displayed above each micro-window, spanning from micro-window 1 (a) to micro-window 4 (d) employing Tikhonov regularization, WACCM6 as the a priori, and filter 4. The observed and fitted lines are represented in blue and green, respectively. The bottom plots shows the residuals (observed minus fitted), with distinct colors indicating different spectroscopy types as labeled. The numbers within the labels denote the root mean square fitting for each micro-window across various spectroscopy databases.

maximize the number of profile comparisons, and considering the relatively stable nature of $CH_4$ (and $N_2O$), we compare FTIR
profiles obtained within a 6-hour time window of the AirCore descent time. The same approach is undertaken when comparing with aircraft observations but the aircraft profiles generally extend up to around 6 km asl altitude whereas the aircore profiles extend up to 20 km or higher.

Figure 6 shows the relative differences in $CH_4$ vertical profiles obtained for each retrieval strategy when compared with AirCore profiles. The sub-panel names correspond to the retrievals listed in Table 2. The individual profile relative differences are displayed in gray, while the mean and median profiles are depicted in red and blue, respectively. To highlight the similarity between AirCore and aircraft profile results, the median relative differences in the troposphere from aircraft profiles are displayed in green in the same figure. The relative differences with aircraft profiles in the troposphere (up to 6 km) are further detailed in Figure 7. Despite the absence of coincidence between AirCore and aircraft profiles, the relative differences exhibit remarkable similarity in both magnitude and shape, enhancing confidence in the assessment of multiple retrievals. It is inter-
esting to observe that changing only the $CH_4$ database spectroscopy leads to a substantial change in the magnitude of profile differences. For example, retrievals r1 and r2 show opposite signs in their relative differences within both the troposphere and





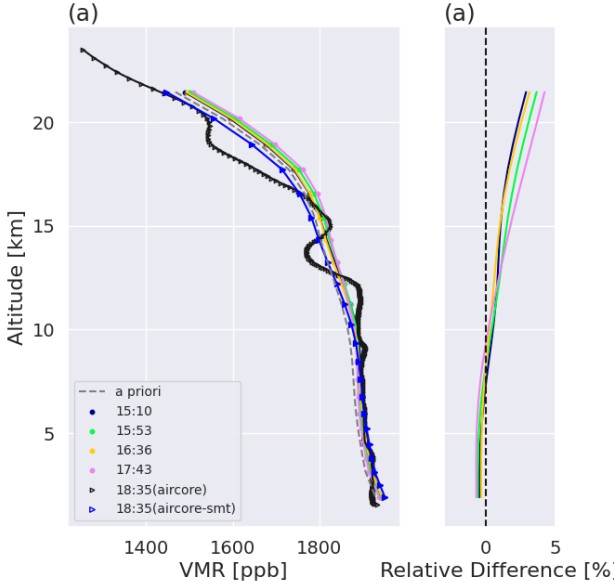

**Figure 5.** (a) Illustrative comparison between FTIR and AirCore $CH_4$ profiles on May 14th, 2019, utilizing ATM20, WACCM6, and Tikhonov regularization (retrieval r4 in Table 2). Various colors denote different times (UT) of HR-FTIR retrievals, with the high-resolution AirCore profile presented in black and the smoothed profile in blue. (b) Relative difference (FTIR minus AirCore) for each corresponding time.

stratosphere, despite having identical retrieval parameters, apart from the spectroscopy. The key distinction between retrievals r1 and r2 is that r1 employs HIT00 and r2 relies on ATM20. Interestingly, using a combination of ATM20 and HIT00 for $CH_4$ yields profile differences that are very similar to those obtained using only HIT00, suggesting that the line parameters for $CH_4$ primarily dictate the profile shape. The inclusion of 3 or 5 micro-windows does not change the shape of the profile difference; however, r5, which uses 5 micro-windows, shows smaller relative differences in both the troposphere and stratosphere compared to r2, which uses the same settings but with only 3 micro-windows. Retrieval strategies showing favorable bias outcomes are encompassed in r7 to r10. Strategy r7 utilizes micro-windows 1, 2, and 3, while r8 employs all five micro-windows, deviating from the micro-windows 1, 3, and 5 recommended by (Sussmann et al., 2011). We observe slightly higher variability in r7, whereas r8 and r9 exhibit more compact variability, particularly in the troposphere. Retrieval r8 is identical to r9, except that r9 uses the OEM approach. Additionally, using monthly mean $CH_4$ a priori profiles does not significantly improve biases, as seen when comparing r10 to r5. It's important to note that past studies have predominantly focused on total column analysis. In such cases, retrieval strategies might produce biases in the troposphere and stratosphere with opposite signs, potentially canceling each other out and reducing the overall bias in total column measurements. In contrast, this work focuses on examining the differences at various altitudes within the vertical profiles, which, due to vertical sensitivity, ultimately lead to distinct biases in tropospheric and stratospheric columns, as well as in total column measurements.

The retrieval methodology implemented in SFIT4 encompasses the vertical profile but is confined in sensitivity to specific altitude ranges. Figure 8a shows typical averaging kernels (row kernels) employed in the $CH_4$ retrieval, while Figure 8b portrays





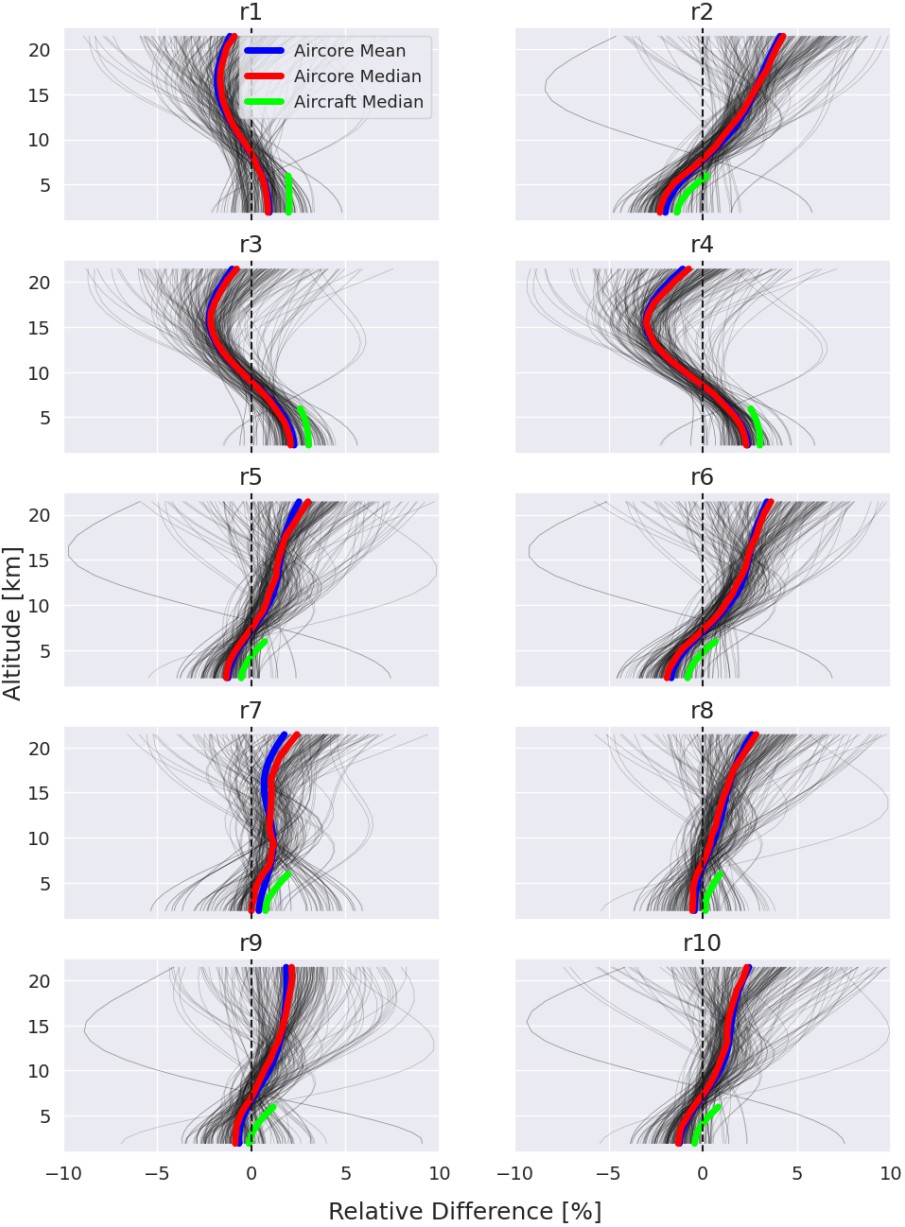

**Figure 6.** Relative differences in CH$_4$ profiles, calculated as the difference between FTIR and AirCore values divided by the AirCore values, employing diverse retrieval strategies. The mean and median profiles are shown in blue and red lines, respectively. The green lines illustrates the relative differences obtained in the lower troposphere using aircraft comparisons, as shown in Figure 7.

the total column averaging kernel and the cumulative sum of DOF, denoting the number of independent layers in the retrieved

profile. Shaded rectangles separate the cumulative sum of DOF, with the first DOF spanning 1.6 - 9.8 km and the second DOF ranging from 9.8 - 22.1 km. Note, that all profile regularization constraints of the retrieval strategies have been adjusted to attain





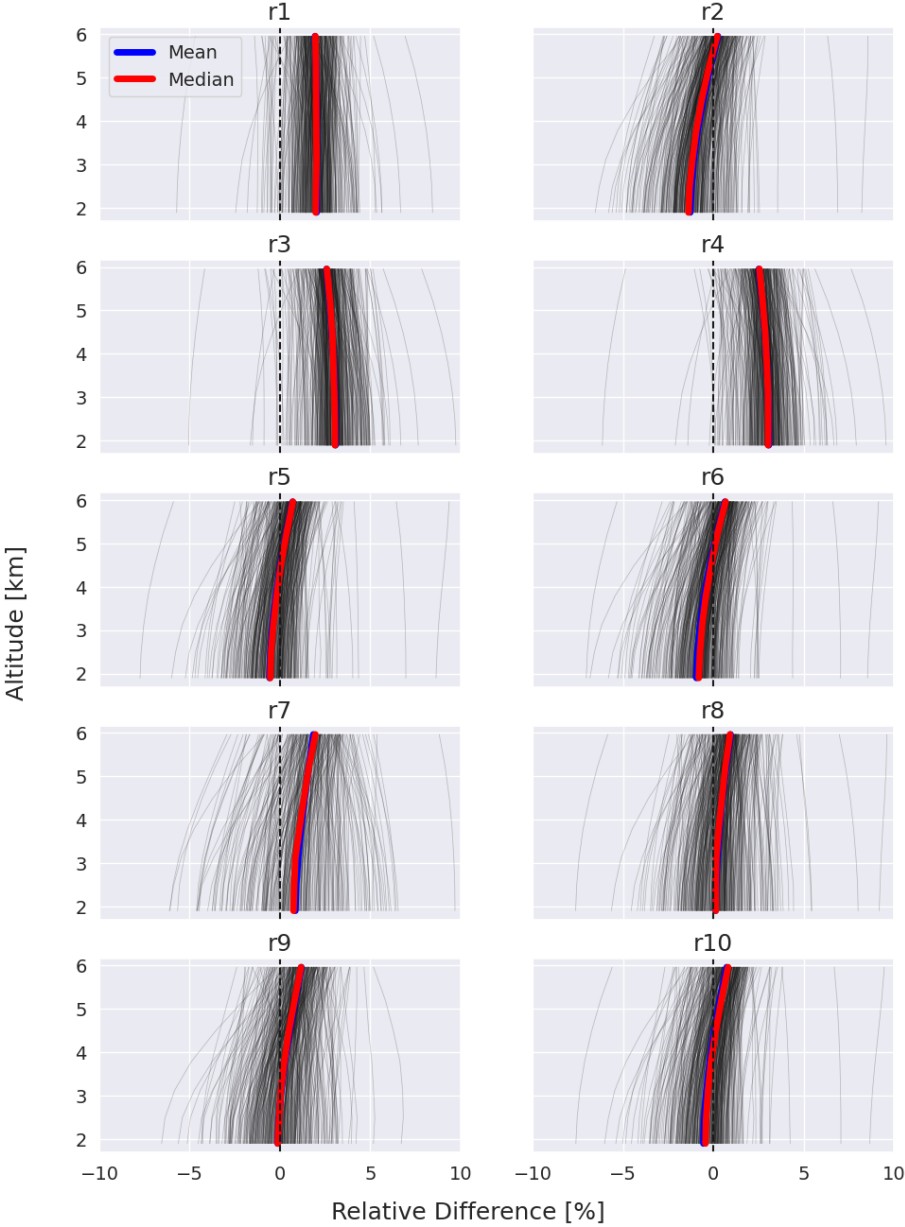

**Figure 7.** Relative difference between FTIR and NOAA aircraft $CH_4$ profiles using several retrieval strategies.

similar DOF, although averaging kernels may differ, particularly between those utilizing Tikhonov and OEM due to the inherent nature of the approaches. For a comprehensive evaluation of these layers, a comparative analysis with AirCore and aircraft data is conducted, emphasizing the mixing ratio weighted by air mass density within the specified altitude ranges mentioned above.

Tables 7 and 8 list the mean and median $CH_4$ relative differences among various retrieval strategies for different partial columns



using AirCore and aircraft profiles, respectively. The last column represents the 95 % confidence interval (CI) calculated. As indicated earlier, tropospheric biases estimated from both AirCore and aircraft observations reveal congruent findings. An integral facet is the assessment of the a priori profile when contrasted with AirCore and aircraft profiles. When comparing WACCM6 across different partial columns, the biases are -1.9 $\pm$ 1.0 % and -1.5 $\pm$ 1.3 % for the troposphere and stratosphere,

respectively. In principle, one might anticipate that retrievals would yield improved and lower biases. Correspondingly, in profile results, r8 demonstrates satisfactory values with mean biases of -0.08 $\pm$ 0.38 % and 0.89 $\pm$ 0.28 % for tropospheric and stratospheric layers, respectively, and 0.39 $\pm$ 0.42 % for aircraft comparisons in the troposphere. Strategy r9, which is identical to r8 but utilizes the OEM approach, produces very similar results.

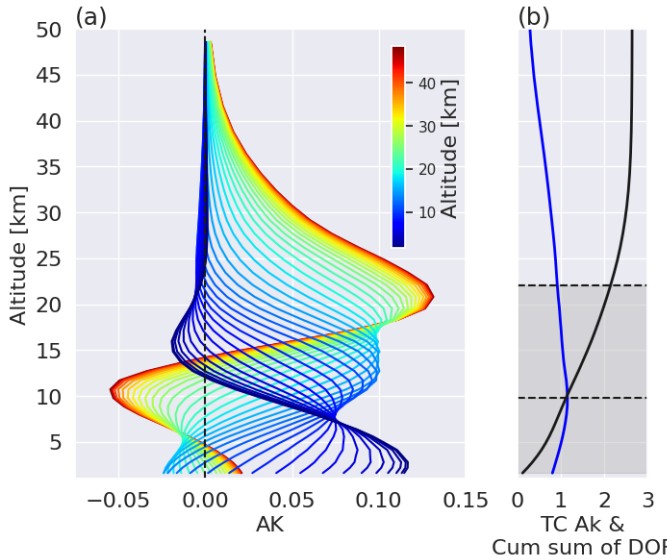

**Figure 8.** (a) Averaging kernel for $CH_4$, representative of November 18th, 2019, at a solar zenith angle of $60.0°$. (b) Total column averaging kernel in blue and cumulative sum of DOF in black. The shaded rectangles indicate the cumulative sum of degrees of freedom (DOF). The first DOF spans 1.6 - 9.8 km, while the second DOF ranges from 9.8 - 22.1 km. This example employs 5 micro-windows, Tikhonov regularization, and WACCM6 as the a priori.

To evaluate $N_2O$, we employ a comparable approach, but use only low to mid tropospheric values from aircraft profiles.

Figure 9 shows the vertical profile differences across all retrieval strategies, and a summary of tropospheric weighted mixing ratio biases is presented in Table 9. In the current retrieval strategy (r1) recommended by the IRWG, there is a systematic positive bias of 1.58 $\pm$ 0.28 % at all tropospheric altitudes. Transitioning to the latest HIT20 spectroscopy (r2) reduces the overall tropospheric bias to 0.12 $\pm$ 0.27 %, although the profile's relative differences shift from negative values below 3 km to positive values above 3 km. Interestingly, employing ATM20 (r3) also results in a systematic positive bias of 1.42 $\pm$ 0.27

%, similar to HIT08. Similar outcomes are observed when using both filters independently. Switching to WACCM6 yields a slight improvement. For instance, in r5, which employs HIT08 with the new WACCM6 profile, the difference is 1.27 $\pm$ 0.28





**Table 7.** $CH_4$ bias results among the different retrieval strategies for the different partial columns using AirCore profiles as the reference.

| Retrieval # | Partial Column [km asl] | Mean bias [%] | Median bias [%] | CI [%] |
|---|---|---|---|---|
| | 1.6 - 10.0 | 0.88 | 0.84 | 0.31 |
| r1 | 10.0 - 20.0 | -1.39 | -1.26 | 0.32 |
| | 1.6 - 20.0 | 0.48 | 0.34 | 0.23 |
| | 1.6 - 10.0 | -0.97 | -1.32 | 0.36 |
| r2 | 10.0 - 20.0 | 1.91 | 1.99 | 0.27 |
| | 1.6 - 20.0 | -0.46 | -0.66 | 0.27 |
| | 1.6 - 10.0 | 1.79 | 1.70 | 0.30 |
| r3 | 10.0 - 20.0 | -1.70 | -1.60 | 0.32 |
| | 1.6 - 20.0 | 1.18 | 1.15 | 0.23 |
| | 1.6 - 10.0 | 1.71 | 1.80 | 0.28 |
| r4 | 10.0 - 20.0 | -2.37 | -2.30 | 0.36 |
| | 1.6 - 20.0 | 0.99 | 1.05 | 0.21 |
| | 1.6 - 10.0 | -0.64 | -0.49 | 0.27 |
| r5 | 10.0 - 20.0 | 1.20 | 1.19 | 0.25 |
| | 1.6 - 20.0 | -0.32 | -0.18 | 0.21 |
| | 1.6 - 10.0 | -0.75 | -0.99 | 0.34 |
| r6 | 10.0 - 20.0 | 1.86 | 1.99 | 0.28 |
| | 1.6 - 20.0 | -0.29 | -0.40 | 0.26 |
| | 1.6 - 10.0 | 0.97 | 0.52 | 0.61 |
| r7 | 10.0 - 20.0 | 0.52 | 1.00 | 0.60 |
| | 1.6 - 20.0 | 0.89 | 0.61 | 0.43 |
| | 1.6 - 10.0 | -0.08 | -0.17 | 0.38 |
| r8 | 10.0 - 20.0 | 0.89 | 0.86 | 0.28 |
| | 1.6 - 20.0 | 0.09 | 0.12 | 0.29 |
| | 1.6 - 10.0 | -0.15 | -0.18 | 0.37 |
| r9 | 10.0 - 20.0 | 1.19 | 1.26 | 0.33 |
| | 1.6 - 20.0 | 0.08 | 0.05 | 0.28 |
| | 1.6 - 10.0 | -0.64 | -0.49 | 0.27 |
| r10 | 10.0 - 20.0 | 1.19 | 1.07 | 0.26 |
| | 1.6 - 20.0 | -0.32 | -0.18 | 0.21 |

%. Further improvement is achieved by applying the Tikhonov approach (r6 and r7) using HIT20. Retrieval r8, combining both filters, demonstrates minimal bias (0.10 ± 0.20 %), suggesting the successful applicability of both filters 3 and 4 across the IRWG/NDACC network. The averaging kernels and cumulative sum of degrees of freedom (DOF) for $N_2O$ are shown in Figure




**Table 8.** $CH_4$ bias results among the different retrieval strategies in the lower troposphere using aircraft profiles as the reference.

| Retrieval # | Mean bias [%] | Median bias [%] | CI [%] |
| --- | --- | --- | --- |
| r1 | 1.90 | 1.92 | 0.32 |
| r2 | -0.76 | -0.79 | 0.40 |
| r3 | 2.89 | 2.88 | 0.34 |
| r4 | 2.84 | 2.84 | 0.31 |
| r5 | -0.26 | -0.10 | 0.36 |
| r6 | -0.47 | -0.43 | 0.39 |
| r7 | 1.08 | 1.24 | 0.64 |
| r8 | 0.39 | 0.30 | 0.42 |
| r9 | 0.35 | 0.41 | 0.41 |
| r10 | -0.24 | -0.22 | 0.37 |

10. The first DOF is captured from the surface to about 8 km, while the second DOF is between 8 to 17 km. Because $N_2O$ appears to be well-mixed among all tropospheric altitudes, the results shown above are representative of the entire troposphere. Unfortunately, long-term stratospheric $N_2O$ data is not yet available in the same fashion as $CH_4$ to evaluate the stratospheric component.

**Table 9.** $N_2O$ bias results among the different retrieval strategies in the lower troposphere using aircraft profiles as the reference.

| Retrieval # | Mean bias [%] | Median bias [%] | CI [%] |
| --- | --- | --- | --- |
| r1 | 1.58 | 1.62 | 0.28 |
| r2 | 0.12 | 0.15 | 0.27 |
| r3 | 1.42 | 1.50 | 0.27 |
| r4 | 0.56 | 0.54 | 0.26 |
| r5 | 1.27 | 1.35 | 0.28 |
| r6 | -0.02 | 0.15 | 0.25 |
| r7 | 0.36 | 0.36 | 0.22 |
| r8 | 0.18 | 0.30 | 0.20 |

## 4.3   Recommended retrieval strategies and error budget

The final suggested retrievals for $CH_4$ and $N_2O$ are presented in Table 10, based on both fitting results using various spectroscopy line parameters and especially by the assessment of vertical profiles and columns using independent AirCore and aircraft profiles. These retrievals correspond to r8 and r9 for $CH_4$ (Table 2), which are identical except that r8 employs Tikhonov regularization while r9 uses OE. For $N_2O$, retrievals follow the r8 strategy ((Table 4). Although Table 2 lists DLR water vapor for $CH_4$ retrievals, we believe that a standardized approach using ATM20 for all gases would provide satisfactory results, while





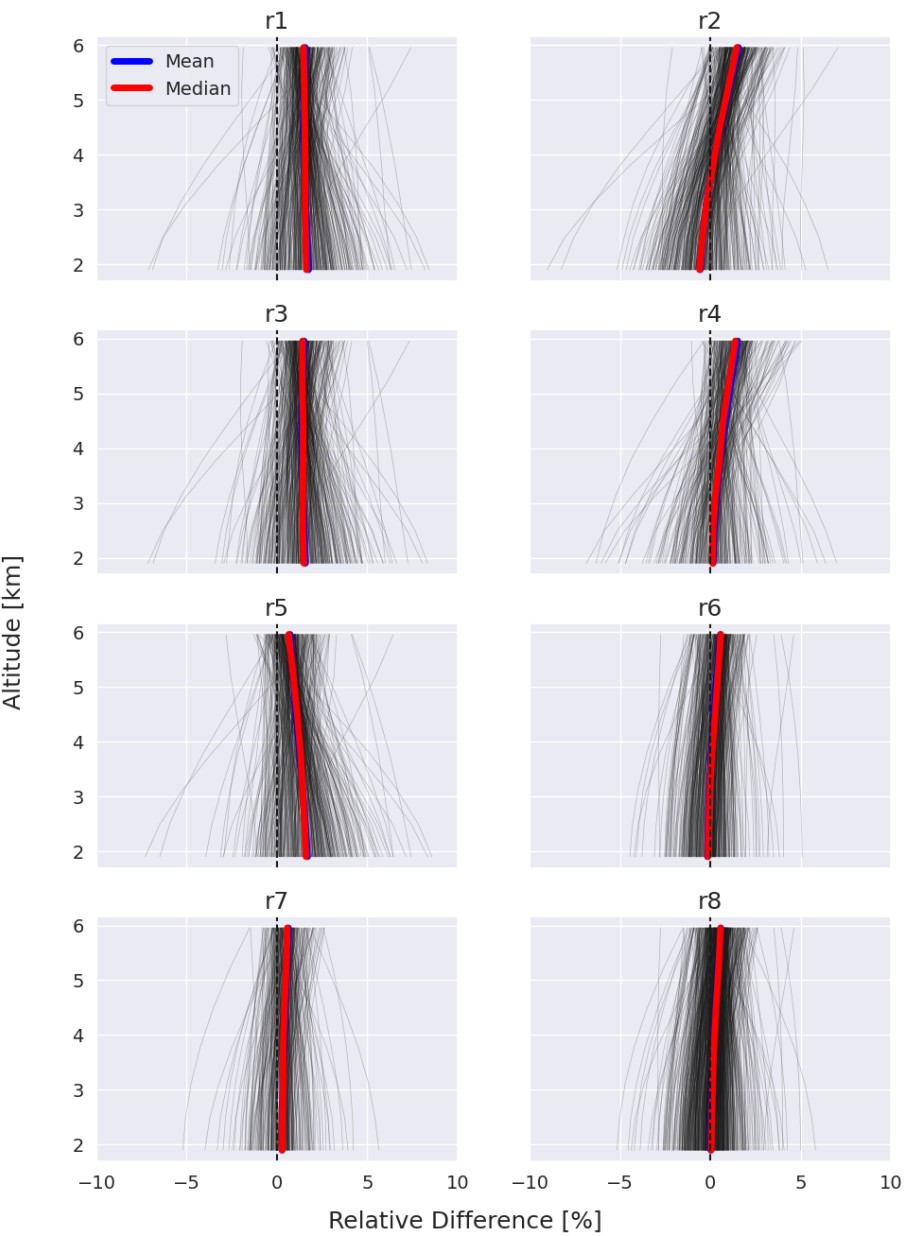

**Figure 9.** Relative difference between FTIR and NOAA aircraft $N_2O$ profiles using several retrieval strategies.

also being more practical and easier to implement on a broader scale. SFIT4 provides uncertainty budgets that combine random and systematic sources following the formalism outlined in Rodgers and Connor (2003). The primary random and systematic error profiles for $CH_4$ and $N_2O$ are illustrated in Figures 11 and 12, respectively. These vertical profile uncertainties are expressed as percentages relative to the mean mixing ratio, with the total respective errors represented by black dotted lines. The



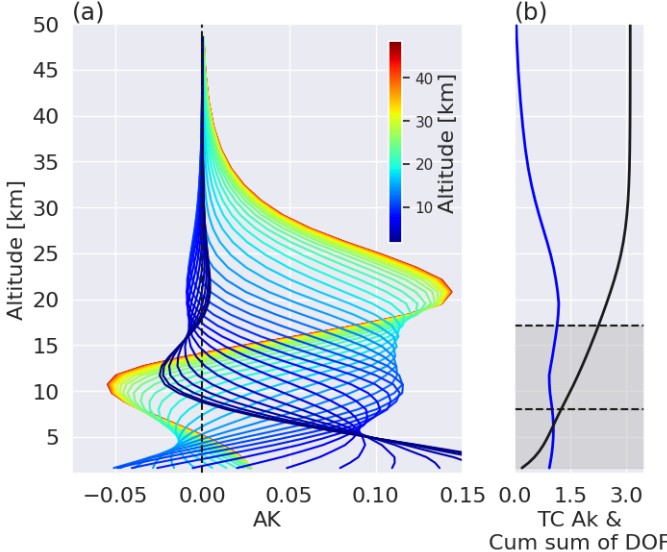

**Figure 10.** Left: $N_2O$ FTIR mean row averaging kernels. Right: total column averaging kernel and cumulative sum of DOF on 20190131.171426 using 5 mw, Tikhonov regularization, and WACCM6 a priori. First DOF is between 1.6 - 8km, second DOF is between 8.0 - 17.12km

error calculation in SFIT4 is detailed in Ortega et al. (2019), with a specific analysis for water vapor at Boulder. The domi-

nant random error in the troposphere for both gases stem from uncertainties in the temperature profile and measurement noise characterized by SNR in the spectral region of interest. Additional error parameters, such as interfering species, apodization function, and solar zenith angle, have a comparatively lesser impact. Major systematic error components arise from absorption line parameters, specifically line intensity ($S$), air-broadened half-width ($\gamma$), and temperature dependence of $\gamma$ ($n$). The lower limit uncertainties reported in HIT20 are used for systematic error retrieval, with values of 5 %, 5 %, and 10 % for $S$, $\gamma$, and

$n$ respectively for $CH_4$, and 2 %, 10 %, and 5 % respectively for $N_2O$. It should be noted that these errors are expected to be significantly smaller for $CH_4$ due to the use of the empirically modified linelist ATM20. Smoothing error is treated separately and excluded from the total error analysis because it is often not well-known and is thus frequently simplified. Tables 11 and 12 provides a summary of mean values of random, systematic, and total uncertainties in the retrieval of both $CH_4$ and $N_2O$, respectively. These values are expressed as percentages in the corresponding partial column as well as the total column errors.

Random error, associated with precision or statistical uncertainty, reflects the inherent variability in measurements and is crucial for meeting precision requirements in observations for $CH_4$, such as growth rate, seasonal cycle amplitudes, inter-hemispheric gradients, among others. This type of uncertainty is typically quantified as the standard deviation or standard error of a set of measurements. For example, (Sussmann et al., 2011) optimized precision to about 0.3 % using 1-sigma standard deviation and diurnal variation in a 7-minute integration, aligning with the retrieval random error of 0.5 % shown in Table 11.





**Table 10.** Suggested $CH_4$ and $N_2O$ final retrieval strategies based on fitting results and assessment with AirCore and aircraft profiles.

|  | $CH_4$ | $N_2O$ |
|---|---|---|
| Micro-windows | 1,2,3,4,5 (Table 1) | 1,2,3,4 (Table 3) |
| Spectroscopy | ATM 2020 and DLR ($H_2O$) | HIT20 |
| A priori | WACCM6 (all main species) | |
| Retrieval constraint | Tikhonov L1 or OEM | |
| IRWG Filter | Filter 3 | Filter 3 and 4 |
| Additional details | Regularization strength optimized for DOF ≈ 2.0-2.5 <br> pre-retrieved $H_2O$, 257 cm OPD <br> Pressure and Temperature NCEP (at least daily) | |

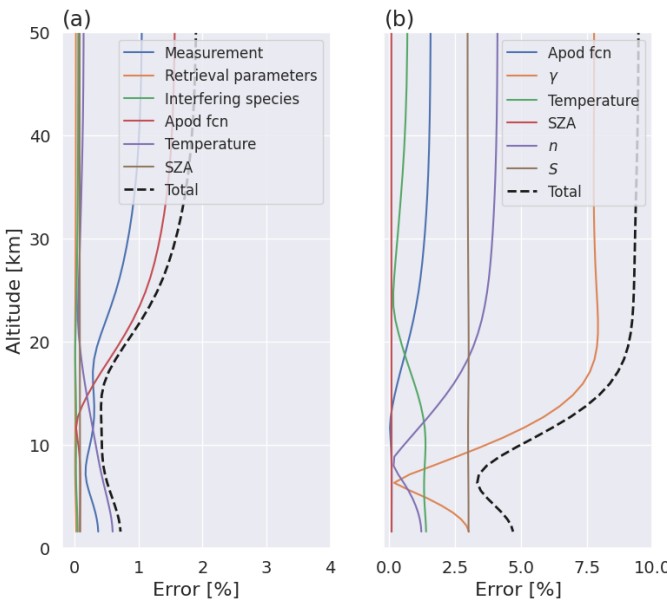

**Figure 11.** Mean vertical profiles of the primary random (a) and systematic (b) uncertainty components in the retrieval of $CH_4$ for the year 2021.

## 4.4 Seasonal variations and trends of $CH_4$: Boulder and Lauder NDACC sites

The retrieval strategies outlined in Table 10 have been applied to long-term measurements from Boulder (2010–2023). To contrast and evaluate the $CH_4$ signal in the Southern Hemisphere, we extended the analysis to the NDACC site in Lauder, New Zealand, located at the National Institute of Water and Atmospheric Research (NIWA) station in Central Otago, at an altitude of 370 m.a.s.l. (Hausmann et al., 2016). FTIR observations at Lauder have been conducted since 1990, but to facilitate comparison with Boulder, we focus on data from 2010 onwards. Between 2010 and 2018, observations at Lauder were collected using a



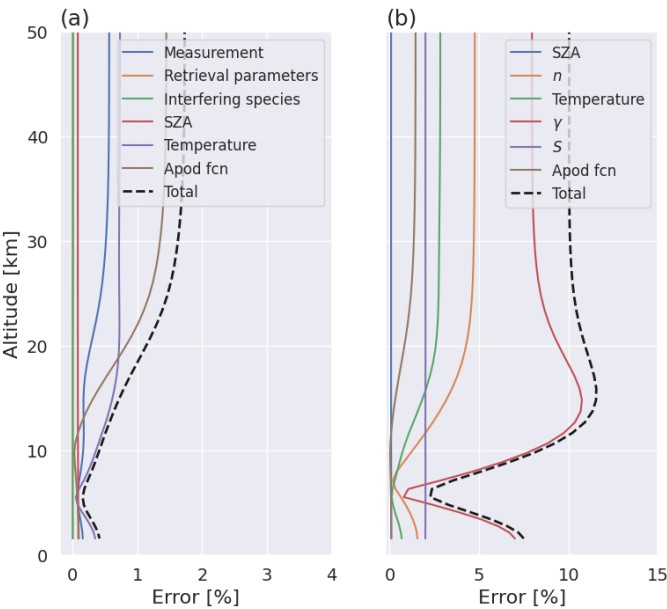

**Figure 12.** Same as 11 but for N$_2$O.

**Table 11.** Mean values of random, systematic, and total uncertainties in the CH$_4$ retrieval for the entire year of 2021 using the final suggested retrieval strategy. These values represent the weighted errors in the corresponding partial column, with the total column errors presented in the bottom row.

| Altitude [km] | Random [%] | Systematic [%] | Total [%] |
| --- | --- | --- | --- |
| 1.6-9.8 | $0.58 \pm 0.06$ | $4.07 \pm 0.41$ | $4.12 \pm 0.41$ |
| 10.7-19.6 | $0.46 \pm 0.03$ | $7.22 \pm 0.47$ | $7.24 \pm 0.47$ |
| Total Column | $0.44 \pm 0.06$ | $3.45 \pm 0.05$ | $3.48 \pm 0.05$ |

Bruker 120HR, while from 2018 onward, a Bruker 125HR has been used. To capture both the long-term trend and seasonal variations, we performed a bootstrap re-sampling analysis on the time series $f(t)$ using the following expression:

$$f(t) = a_0 + a_1(t - t_0) + \sum_{n=1}^{N} b_n \cos(n\pi t) + \sum_{n=1}^{N} c_n \sin(n\pi t) \tag{1}$$

Here, the first two terms represent the linear trend, where $a_0$ is the intercept, $a_1$ is the slope (indicating the long-term
trend), and $t_0$ is the time of the initial observation. The third and fourth terms are a Fourier series that models the seasonal variability, with a second-order Fourier series ($N = 2$) adequately capturing the seasonal cycle. The bootstrap analysis was conducted $5 \times 10^3$ times, and the mean slope was used to quantify the annual rate of change, while the standard deviation provided an estimate of the uncertainty. An advantage of NDACC's high-resolution FTIR measurements is that these retrievals yield 2 DOFs, allowing us to distinguish between the tropospheric and stratospheric CH$_4$ columns. Figure 13 displays the



**Table 12.** Mean values of random, systematic, and total uncertainties in the $N_2O$ retrieval for the entire year of 2021 using the final suggested retrieval strategy. These values represent the weighted errors in the corresponding partial column, with the total column errors presented in the bottom row.

| Altitude [km] | Random [%] | Systematic [%] | Total [%] |
| --- | --- | --- | --- |
| 1.6-9.8 | $0.29 \pm 0.09$ | $5.21 \pm 0.85$ | $5.22 \pm 0.85$ |
| 10.7-19.6 | $0.66 \pm 0.13$ | $10.74 \pm 0.75$ | $10.76 \pm 0.75$ |
| Total Column | $0.16 \pm 0.05$ | $2.58 \pm 0.09$ | $2.59 \pm 0.09$ |

time series of $CH_4$ tropospheric, stratospheric, and total columns at Boulder and Lauder, derived using the retrieval strategies recommended in this study. The altitude ranges for the tropospheric and stratospheric columns extend from the site altitude to the altitude where the DOF is closest to one, and from that altitude up to 25 km, respectively. For Boulder, this altitude is 9.3 km, and for Lauder, it is 7.6 km. While our goal is to differentiate the tropospheric and stratospheric signals, we opted not to exclude the tropopause in this analysis, though this might be an important factor for more detailed assessments in future

studies. To enhance clarity, only data points within 5 % of the seasonal modulation's amplitude are shown in gray. Monthly means are represented by circles with associated standard deviations, while solid lines represent the seasonal modulation and trend components derived from the bootstrap analysis. The annual rate of change for each column, relative to the mean of the full time series, is presented in Table 13. Another important assessment is the characterization of the seasonal variation. For a closer look at seasonal variation, we removed the linear trend using:

$$f(t)_d = f(t) - (a_0 + a_1 (t - t_0)) \tag{2}$$

The seasonal variation is then represented by the monthly means of this de-trended data, along with their uncertainties, in Figure 14 for both Boulder and Lauder over the period 2010–2023. This analysis is not intended to validate the trends or seasonal variations but to present the data, serving as a basis for future studies aiming to investigate these aspects in more detail. For additional insights into the comparison between various retrievals and AirCore and surface in situ trends and seasonal

variation, see supplemental information. Overall, the linear trends and seasonal variations remain consistent across different retrieval strategies, with only the bias varying, as demonstrated in the comparisons with AirCore and aircraft measurements.

**Table 13.** Linear trend analysis of $CH_4$ time series at Boulder and Lauder for tropospheric, stratospheric, and total columns as shown in Figure 13.

| Column | Boulder [%/year] | Lauder [%/year] |
| --- | --- | --- |
| Tropospheric | $0.702 \pm 0.007$ | $0.510 \pm 0.005$ |
| Stratospheric | $0.390 \pm 0.012$ | $0.715 \pm 0.009$ |
| Total | $0.588 \pm 0.005$ | $0.581 \pm 0.004$ |



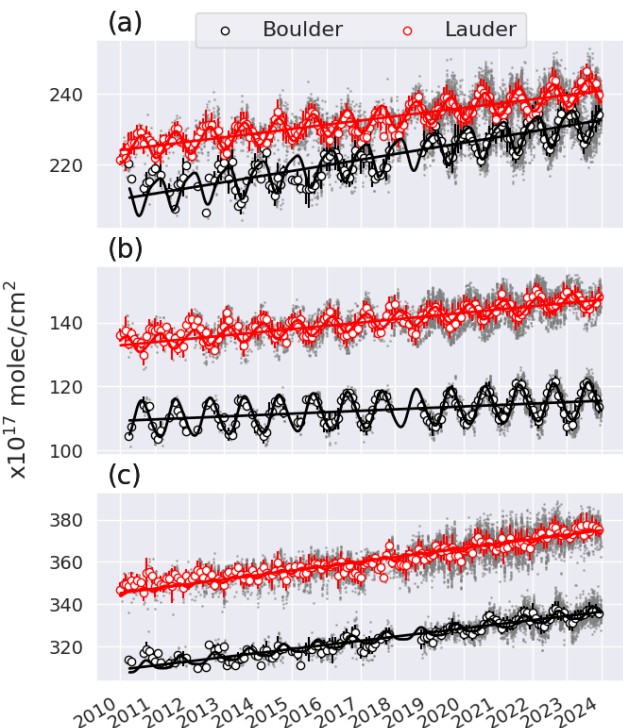

**Figure 13.** Time series of CH$_4$ columns at Boulder (black) and Lauder (red) derived using the suggested retrieval strategy, separated into (a) tropospheric, (b) stratospheric, and (c) total columns. Gray dots represent individual data points as described in the main text, circles denote monthly mean values, and the solid lines show the combined seasonal modulation and trend component derived from the entire dataset.





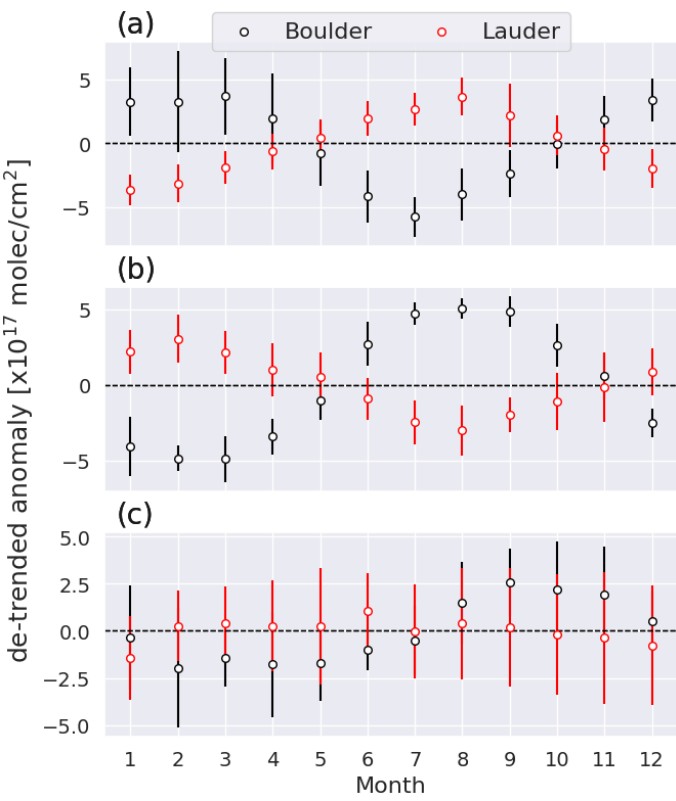

**Figure 14.** Monthly mean values of CH$_4$ derived from the detrended analysis are presented for Boulder (black) and Lauder (red) across the troposphere (a), stratosphere (b), and total columns (c). Error bars represent the standard deviations of the multi-annual monthly means.



## 5    Conclusions

Over recent years, there has been noticeable variability in the retrieval strategies for CH$_4$ among different ground-based FTIR instruments within the IRWG of NDACC. Acknowledging advancements in spectroscopy and the need for retrieval consistency

among the IRWG of NDACC, our study evaluates various retrieval strategies, encompassing not only CH$_4$ but also N$_2$O. Using observations from Boulder, Colorado, we initially assess fitting results across diverse spectroscopy databases, including the latest HITRAN databases, the ATM spectroscopy utilized by TCCON and IRWG, a combination of them, and a water vapor line list from DLR. We also evaluate previously identified suitable micro-windows. While some cases exhibit clear improvements in residuals for certain spectroscopy, in other instances, residual differences are less distinct, despite significant

variations in profile retrievals. Thus, we emphasize the importance of the subsequent step: comparing independent long-term AirCore and aircraft insitu profile observations with the various retrieval strategies, i.e., 10 different retrieval strategis for CH$_4$ utilizing different micro-windows, spectroscopy, regularization approach (Tikhonov or optimal estimation) and 8 retrievals for N$_2$O mainly focusing on spectroscopy, regularization approach, and a priori profiles. Notably, both AirCore and aircraft profiles exhibit remarkable similarity in their relative differences compared to all FTIR retrievals, strengthening our confidence

in the assessment. Moreover, bias analyses across various atmospheric layers for both CH$_4$ and N$_2$O contribute to refining retrieval methodologies. Clear biases were observed when comparing with independent profiles, making the exclusion of certain retrievals straightforward. We propose optimized retrieval settings for CH$_4$ and N$_2$O, leveraging a first-order Tikhonov method and incorporating a priori profiles from the latest WACCM model simulations to enhance accuracy. While the HITRAN 2020 spectroscopic database is effective for N$_2$O, ATM 2020 provides better results for CH$_4$, with slight improvement observed

when paired with the water vapor line list from DLR; however, this enhancement may be site-dependent. Furthermore, profile comparisons reveal biases within different atmospheric layers.For the proposed retrievals, we obtain a mean bias of -0.08 ± 0.38 % and 0.89 ± 0.28 % for the tropospheric and stratospheric layers of CH$_4$ using AirCore, respectively, and 0.39 ± 0.42 % for aircraft comparisons in the troposphere. For N$_2$O, the bias in the troposphere is approximately 0.18 ± 0.2 %. Furthermore, we provide uncertainty budgets, including random and systematic sources, to comprehensively understand

error sources, guiding future refinement efforts. Random errors of about 0.5 % are found, mainly attributed to temperature profile uncertainties and measurement noise, dominate in the troposphere for both gases. Overall, our findings significantly contribute to advancing understanding of atmospheric composition and advocate for an effective harmonized approach among all IRWG/NDACC sites. The accuracy of our retrievals allows us measurements of growth rates, seasonal cycle amplitudes, interhemispheric gradients, and supports satellite mission characterization. Although Boulder benefits from unique long-term

AirCore and aircraft profile observations, these capabilities are not widespread across IRWG/NDACC sites, underscoring the importance of deploying similar observations systems elsewhere in the NDACC network.

*Code and data availability.* The profile retrieval algorithm, SFIT4 (Hannigan et al., 2024), is freely available and can be obtained at https://wiki.ucar.edu/display/sfit4/SFIT+Core+Code. The Level 2 NOAA AirCore dataset (v20230831) (Baier et al., 2021) is publicly available via the NOAA data repository: https://doi.org/10.15138/6AV0-MY81. Atmospheric observations from the NOAA GML Carbon Cycle Aircraft



Vertical Profile Network (Sweeney et al., 2015; **?**) can be obtained at https://gml.noaa.gov/ccgg/aircraft/. Additional data used in this study can be provided upon request from the corresponding author.

*Author contributions.*  IO and JH conceptualized the project. IO carried out the data analysis, generated all figures and plots, and prepared the initial draft of the manuscript. BB, KM, and DS contributed the AirCore, aircraft, and Lauder FTIR datasets, respectively, and provided substantial input throughout the study. All co-authors contributed to discussions, offered feedback on the manuscript, and assisted in its
preparation.

*Competing interests.*  The authors declare no competing interests.

*Acknowledgements.*  This material is based upon work supported by the National Center for Atmospheric Research (NCAR), which is a major facility sponsored by the National Science Foundation under Cooperative Agreement No. 1852977. This study has been supported under contract by the National Aeronautics and Space Administration (NASA) award No. NNX17AE38G. This study was supported by
NOAA Climate Program Office's Atmospheric Chemistry, Carbon Cycle, and Climate program, grant number NA23OAR4310285. We thank the many people at NOAA GML that contributed to the AirCore and aircraft datasets used in this study, including Andy Crotwell, Molly Crotwell, Ed Dlugokencky, Jack Higgs, Xin Lan, Pat Lang, Thomas Legard, Monica Madronich, Eric Moglia, John Mund, Don Neff, Tim Newberger, Colm Sweeney, Pieter Tans, and Sonja Wolter.



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
