# Peer review of "Advancing $CH_4$ and $N_2O$ retrieval strategies for NDACC/IRWG FTIR observations with the support of airborne in situ measurements"

_EGUsphere, 2024_

## Author Comment (AC1)

Black: Referee's comments
Blue: Author's reply

We sincerely appreciate the Referee's time and effort in reviewing our manuscript. Below, we provide a point-by-point response to the comments.

This manuscript by Ortega et al. presents an extensive evaluation of retrieval strategies for two major greenhouses gases, namely, methane and nitrous oxide, applied to high resolution solar absorption spectra obtained with Fourier transform infrared (FTIR) spectrometers.

For both targets, the authors carefully investigate the effect of the line parameters selection and of the adopted regularization on the retrieved geophysical products, using the SFIT4 algorithm. The parametrization is tuned such as to reach comparable information content (degree of freedom for signal or DOF of 2-2.5). Observations obtained at Boulder, CO, in the framework of the NDACC network are the focus here. Indeed, the authors capitalize on concurrent airborne (air core or aircraft) in situ measurements available in the vicinity of this site over several years.

This way, the numerous strategies investigated can be ranked not only on the basis of the smallest spectral residuals achieved, but also, and importantly, by minimizing the bias between the remote-sensing and in situ data.

The inter-technique bias for CH4 and N2O are determined considering large samples and found small or even not significant. Uncertainty budgets are also evaluated and presented considering the prescribed strategies.

Overall, the results are appropriately presented and discussed, and the manuscript reads well despite the large number of figures and tables.

Major comments

This work will undoubtedly be very useful for the NDACC FTIR community, likely leading to better harmonization and consistency across the stations while potentially minimizing biases with other techniques, consolidating the resulting data sets, making of them more reliable and relevant ensembles for satellite and atmospheric model validation.

Still, this manuscript remains very technical by nature and possibly of limited relevance for the broader scientific community. In my view, it could be more appropriate to consider this work as a "Technical note" and to include in a supplement the detailed descriptions of the selected and recommended retrieval strategies, including the regularization (especially for the optimal estimation approach). This way, this paper would be optimized for its most likely end users.

We appreciate the reviewer's recognition of the value of this work for the NDACC FTIR community and its potential impact on data harmonization, bias minimization, and broader applications in satellite and model validation. Given the extensive analysis of retrieval strategies, uncertainty assessment, and validation against independent in situ measurements, we believe that AMT is an appropriate venue for this study. We do not believe adding the phrase 'Technical note' is required as it is implicit within the journal scope and focus. While the results are highly relevant for NDACC, they also extend beyond this network by providing insights applicable to other FTIR retrieval efforts, satellite validation studies, and atmospheric modeling. The methodology and findings contribute to the broader atmospheric science community by improving the accuracy and consistency of ground-based remote sensing techniques for $CH_4$ and $N_2O$, which are critical for understanding greenhouse gas trends.

Finally, I found the section on the trend analysis quite irrelevant in the present context, considering that this study aims at the determination of optimum FTIR products as part of a network effort. Moreover, this section suffers from flaws and does not bring very useful information to the reader. More specifically:

- there is no information about the tool or method that is used to derive the trends, what is the approach used to estimate the trends and their associated uncertainties?
- the statistical uncertainties are extremely small: are they really representative and robust?
- there are some dissimilarities among the NH and SH trends in the stratosphere (see Table 13): this might be at least partly related to well-known stratospheric asymmetries that were the subject of earlier studies for other long-lived tracers (e.g., by Strahan et al., 10.1029/GL088567, 2020), but the authors do not discuss neither comment this feature
- furthermore, the tropospheric trends are also statistically different; can we really expect this for such a well-mixed greenhouse gas?
- the CH4 trend is evaluated, but not the N2O one; this is not explained nor justified

My suggestion would therefore be to just remove the trend section (section 4.4) and to investigate the CH4 and N2O trends in a follow-up paper, involving a larger number of FTIR stations after network-wide implementation of the recommended strategies.

We appreciate comments/feedback on the trend analysis section. Given these valid points (and also note by Reviewer #2), we agree that trend analysis is beyond the scope of the current manuscript. Thus, we have removed Section 4.4 and will consider investigating $CH_4$ and $N_2O$ trends in a dedicated follow-up study, incorporating a larger number of FTIR stations after the recommended retrieval strategies are implemented network-wide.

Specific comments and suggestions

An originality of this manuscript is the use of concurrent in situ data to assess FTIR retrieval strategies. I would suggest mentioning that strength in the title, perhaps by adding "FTIR observations with the support of airborne in situ measurements".

We have incorporated the suggestion and the title now is: "Advancing $CH_4$ and $N_2O$ retrieval strategies for NDACC/IRWG FTIR observations with the support of airborne in situ measurements"

Line 36: its relatively short atmospheric…

Done

Lines 88-89: it might be relevant to provide information about the filters 3 and 4? What are their characteristics and respective advantages or limitations? Or provide a reference?

We have provided additional details on the filters. The revised paragraph is as follows:

*"Optical bandpass filters are employed to optimize the signal-to-noise ratio (SNR) within the near and mid-infrared spectral range. Hannigan et al. (2009) provide a list of typical band limits for each NDACC filter, along with the gases exhibiting absorption features and typically retrieved, covering a spectral range of 750–5000 $cm^{-1}$. For $CH_4$, NDACC filter 3 is used, while for $N_2O$, both filters 3 and 4 can be used, though filter 3 is predominantly used."*

Line 90: maximum optical path difference

Done

Line 124: to a common altitude scheme…

Done

Figure 2.a. why is the methane scale in ppm here, and in ppb previously? Perhaps harmonize?

Done

Table 5 and similar: would bolding the best results helps the reader to identify the most relevant strategy?

Thank you for the recommendation; we have implemented this suggestion.

Line 308: remove double opening parenthesis

Done

Figure 11 (and 12): is it relevant to show the altitude range above 30-35 km, where there is no information available?

Thank you for the suggestion. While we acknowledge that there is limited information above 35 km, we have chosen to keep the axis limit at 50 km as there is not much difference in the results between 35 km and 50 km, and the legend remains more legible in the current layout.

Line 331: can we consider that 13 years of data provide a "long-term" view for a geophysical parameter?

As suggested above, we have removed this section.

Figure 12: are the uncertainties similar for "filter 3" and filter 4"?

Yes, uncertainties are very similar. In the updated manuscript we have mentioned that the uncertainty reported in table/figure are for both filter 3 and filter 4.

Figure 13 (and 14): I found counterintuitive to place the panel for the stratosphere below the one for the troposphere; also and of kept, they could be enlarged for readability

As suggested above, we have removed this section.

Line 395: remove the question mark after "Sweeney et al., 2015"?

Updated: The latest manuscript includes an additional reference that was previously omitted in the earlier version.

---

## Author Comment (AC2)

Black: Referee's comments
Blue: Author's reply

We sincerely appreciate the Referee's time and effort in reviewing our manuscript. Below, we provide a point-by-point response to the comments.

In this manuscript, the authors present a broad evaluation of different retrieval strategies for methane ($CH_4$) and nitrous oxide ($N_2O$)—two major greenhouse gases that also affect the ozone layer—from solar absorption infrared spectra measured at the NDACC (Network for the Detection of Atmospheric Composition Change) station in Boulder, Colorado, USA.

The investigated retrieval strategies make use of different molecular absorption databases (including the latest releases), a priori profiles, spectral windows, and regularization methods, thus providing an exhaustive evaluation. The comparison is based on the assessment of fitting residuals (attributed to spectroscopic parameters) as well as the comparison of retrieved profiles and partial columns to a unique dataset from AirCore ($CH_4$ only) and aircraft measurements (both $CH_4$ and $N_2O$).

These investigations ultimately lead to recommended retrieval strategies for the Infrared Working Group (IRWG) of NDACC, aiming to harmonize high-quality global datasets of these critical trace gases.

Given that this manuscript presents novel methods for producing high-quality datasets of two major greenhouse gases and leverages a unique dataset of airborne in situ measurements to validate these methods, I recommend its publication in Atmospheric Measurement Techniques once my comments have been addressed.

Major comments

$N_2O$ retrieval strategy

Based on the results presented in the manuscript, the $N_2O$ retrieval strategy that fits both filter 3 and filter 4 spectra simultaneously is claimed to be the best among the tested strategies and is therefore recommended by the authors. I have several comments regarding this conclusion:

- Fitting multiple spectra simultaneously is not a common approach within the IRWG and raises methodological questions. The manuscript does not provide sufficient details on how this retrieval strategy is implemented in practice. For example, were any time co-location criteria applied when selecting the spectra? FTIR high-resolution spectra have a long integration time (several minutes), and it is common to record several spectra for the same filter before switching to another one. If filter 3 is not immediately followed by filter 4 in the measurement protocol, then the selected spectra could span a relatively long time. What, then, is the impact of varying solar zenith angles and other parameters? Additionally, is the assessed error budget comparable

to other strategies, such as r6? Since this strategy is recommended for implementation, more details should be included in the manuscript to help network members adopt it in the future.

We did not intend to suggest that the retrieval fits use filters 3 and 4 simultaneously. Instead, the retrievals are performed independently for each filter, but the results are combined to maximize the number of retrievals. The sentence explaining this now reads as follows:

*"Additionally, two optical band pass filters, denoted as filters 3 and 4 within the IRWG, are available for utilization in the $N_2O$ retrieval process. In this case, we conducted separate $N_2O$ retrievals using each filter independently, as well as a retrieval approach where both filters were used independently, but the results were combined without distinguishing between filters."*

- Looking at Figure 9 and Table 9, one could argue that the retrieval strategy r6 yields comparable or even better results than r8 (the recommended strategy). Since r6 is essentially an updated version of the existing IRWG strategy (incorporating HITRAN 2020 instead of 2008, a new version of the WACCM profiles, and a shift to Tikhonov L1 regularization), wouldn't it be simpler for the network to implement r6 instead?

As the reviewer noted, incorporating HITRAN 2020 instead of 2008, along with the updated WACCM profiles, significantly improves the retrievals. To maximize the number of retrievals, sites can utilize results from either filter 3 or 4, as the differences remain small within the confidence interval. Using filter 3, filter 4, or a combination of both would be a straightforward approach for all sites. Please find the revised text in the manuscript, here it is a paragraph that is revised:

*"The combination of HIT20 and WACCM6 further improves the comparison, as seen in retrievals r6 (-0.02 ± 0.25 %) for filter 3 and r7 (0.36 ± 0.22 %) for filter 4, both employing the Tikhonov approach. Retrieval r8, which uses both filters, exhibits minimal bias (0.18 ± 0.20 %), demonstrating the feasibility of using filters 3 and 4 across the IRWG/NDACC network. Additionally, combining both filters maximizes the number of available retrievals. The OEM retrieval approach was also tested with HIT20 and WACCM6, yielding similar results to the Tikhonov approach. Specifically, the OEM-based retrieval showed a bias of 0.02 ± 0.23 %, which falls within the uncertainty of r8 using Tikhonov."*

Trend analysis section

I understand that the purpose of the trend section is not to conduct a full trend analysis but rather to provide a foundation for future studies. However, I would argue that the residual resampling bootstrap method is not well-suited for geophysical time series. This method is known to be sensitive to the strong autocorrelation typically observed in such datasets, often leading to underestimated confidence intervals for the slope parameter. This issue should at least be acknowledged in the manuscript.

As currently presented, this section does not contribute significantly to the manuscript and may be outside its primary scope and objectives. The authors might consider either improving the methodology—potentially by comparing explicitly their trends with those derived from independent datasets—or removing this section altogether.

We appreciate comments/feedback on the trend analysis section. Given these valid points (and also note by Reviewer #1), we agree that trend analysis is beyond the scope of the current manuscript. Thus, we have removed Section 4.4 and will consider investigating CH4 and N2O trends in a dedicated follow-up study, incorporating a larger number of FTIR stations after the retrieval strategies are implemented network-wide.

Minor comments/suggestions/typos

Page 10: The manuscript demonstrates that including HI00 for $CH_4$ spectroscopic parameters results in the lowest residuals in window 4 and, to a lesser extent, in window 5. However, as shown later in Section 4.2, the r8 and r9 (ATM20) strategies provide the best comparison to the in situ data. What is causing better results with higher residuals is probably not known, but could this be discussed or acknowledged somewhere in the paper? Perhaps in Section 4.2 or the conclusion section?

Indeed, as discussed in Section 4.1 (Inter-comparison among spectroscopy databases), while the choice of spectroscopy database is important, the assessment of profiles using independent observations is also important. As noted in the comment, HIT00 performs slightly better for MW4 and MW5 but does not outperform ATM in other micro-windows. A key takeaway is that despite recent updates, HITRAN still requires refinements for $CH_4$. Accordingly, we have included/modified the following sentence in the conclusions:

*"While the HITRAN 2020 spectroscopic database is effective for $N_2O$, it exhibits suboptimal residuals for $CH_4$, indicating the need for further refinements. In contrast, ATM 2020 yields better results for $CH_4$, with a slight improvement when combined with the DLR water vapor line list. However, this improvement may be site-dependent."*

Page 10, the sentence in lines 219 to 220 is not clear. There is a typo with the molecular absorption databases: HIT04CH4 should be HIT00CH4. For the statement: "micro-window 4 shows superior residuals", I interpret it as larger residuals, while it is the opposite.

Done. You are correct, we intended to refer to lower residuals. The sentence has been revised accordingly.

P2, L37: add the actual GWP for CH4

Done

P3, L79: define semi-co-located?

*Since this is just and an introductory sentence, we have revised the sentence as follows:*

*"Boulder offers a unique opportunity with its extensive, multi-year coverage of AirCore, aircraft, and IRWG/NDACC profiles, which sample $CH_4$ in close proximity, allowing for the integration of numerous observations into our analysis."*

P3, L89: define spectral range of filters

*We have provided additional details on the filters. The revised paragraph is as follows:*

*"Optical bandpass filters are employed to optimize the signal-to-noise ratio (SNR) within the near and mid-infrared spectral range. Hannigan et al. (2009) provide a list of typical band limits for each NDACC filter, along with the gases exhibiting absorption features and typically retrieved, covering a spectral range of 750–5000 $cm^{-1}$. For $CH_4$, NDACC filter 3 is used, while for $N_2O$, both filters 3 and 4 can be used, though filter 3 is predominantly used."*

Section 2.2, this is only a suggestion: add subsections for AirCore (2.2.1) and aircraft (2.2.2)?

*We have adopted the suggestion.*

P4, L116: Maybe add that those years (2018 to 2022) were selected because they are spanning the common period with aircraft data?

*We have modified the sentence as follow:*

*"Our investigation includes profiles collected from 2018 to 2022, enabling the evaluation of different retrieval strategies using a substantial set of coincident FTIR, AirCore, and aircraft observations."*

Section 4.2, again this is only a suggestion: To improve readability and help the reader navigate the results more easily, this section could be further divided into two subsections, one for $CH_4$ and one for $N_2O$. The first paragraph (lines 235–247) could remain as it is, followed by the introduction of Section 4.2.1 at line 248 (dedicated to $CH_4$) and Section 4.2.2 at line 289 (dedicated to $N_2O$). This structuring might make it easier to follow the discussion of each gas separately.

*We have implemented this suggestion in the revised manuscript.*

P12, L244: remove the brackets for "(and N2O)"

*Done*

P15, L279: I am not sure to understand how the partial columns were estimated for the in situ data. This sentence could be clarified.

We have added a reference and revised the sentence accordingly. Within the paragraph we have included this:

"*To comprehensively evaluate these layers, we conduct a comparative analysis with AirCore and aircraft data, using mixing ratios weighted by air mass density within the specified altitude ranges (see Ortega et al. (2021) for details on weighted mixing ratios).*"

P16, L283-285: I am not sure to understand this sentence. What exactly is being compared here? These numbers are not included in any tables.

The objective here is to use the a priori profile from WACCM simulations and compare it with airborne observations. This serves as an additional analysis, where improved agreement with the retrievals is expected. Indeed, as reported, the biases in the retrievals are reduced compared to those in the model simulations. We agree that this sentence was not very clear, the following sentence has been revised as follows:

"*Strategy r8 demonstrates satisfactory performance, with mean biases of -0.08 ± 0.38% and 0.89 ± 0.28% for the tropospheric and stratospheric layers, respectively, and 0.39 ± 0.42% for aircraft comparisons in the troposphere. Strategy r9, which is identical to r8 but employs the OEM approach, produces very similar results. Additionally, we assessed the a priori profile by comparing it with AirCore and aircraft profiles. When evaluating WACCM6 across different partial columns, the biases are -1.9 ± 1.0% and -1.5 ± 1.3% for the troposphere and stratosphere, respectively. Retrievals yield improved and lower biases, with r8 and r9 showing significant reductions.*"

P17: there is a typo for the bias, it is 0.18 instead of 0.10 (see Table 9)

Corrected.

P18, L302. Even if I understand that the authors are referring to the AirCore technique, one could argue that long-term satellite datasets exist (e.g., ACE-FTS). I would re-phrase or remove this statement.

Reviewer is correct, we are referring to AirCore and the long-term comparison. We have this updated sentence:

"*Unfortunately, long-term stratospheric AirCore $N_2O$ data is not yet available in the same fashion as $CH_4$ to evaluate the stratospheric component using the same approach.*"

In the first or second sentence of the conclusion, I would suggest highlighting that this study leverages a unique airborne in situ dataset to perform a profile comparison—an approach that is rarely undertaken within the IRWG. This aspect significantly strengthens the study's findings and underscores its contribution to improving retrieval strategies.

We have revised the conclusions as suggested and a part of the conclusion is as follows:

"*Using observations from Boulder, Colorado, we initially assess fitting results across diverse spectroscopy databases, including the latest HITRAN databases, the ATM spectroscopy utilized by TCCON and IRWG, a combination of them, and a water vapor line list from DLR. We also evaluate previously identified suitable micro-windows. While some cases exhibit clear improvements in residuals for certain spectroscopy, in other instances, residual differences are less distinct, despite significant variations in profile retrievals. Thus, we emphasize the importance of the subsequent step: comparing independent long-term AirCore and aircraft insitu profile observations with the various FTIR retrieval strategies. By utilizing this dataset, we strengthen the evaluation of various retrieval strategies for $CH_4$ and $N_2O$, an approach rarely undertaken within the IRWG.*"

P26, L388-389: this is a strong conclusion, but it unfortunately lacks supporting references. As the trend section is currently conducted and written, it does not fully support this statement. I suggest rephrasing it to make it less definitive, ensuring that this conclusion aligns with the evidence presented in the manuscript.

This sentence has been removed, as Section 4.4 ("Seasonal Variations and Trends of CH4: Boulder and Lauder NDACC Sites") has been removed in the revised manuscript.

---

## Author Response (AR2)

Black: Editor's comments
Blue: Author's reply

We appreciate the Editor's additional comments and feedback. Below, we provide our response.

The authors have revised the manuscript according to the referees' suggestions. I have one minor additional comment that I suggest incorporating before it is published - specifically to provide an actual value for the temporal and spatial co-incidence of the AirCore with the ground-based FTIR data, somewhere presumably in Section 2.2. I don't think this affects any conclusions, but it would be useful for the reader to know this.

After including this information, the manuscript should be published.

Some of this information is already included in Section 4.2 (Inter-comparison with AirCore and aircraft profiles); however, we have expanded the description. The revised paragraph is as follows:

*"To maximize the number of profile comparisons, and considering the relatively stable nature of $CH_4$ and $N_2O$, we compare FTIR profiles obtained within a 6-hour time window of the AirCore descent time. The mean distance between the AirCore descent location and the FTIR site is 76.8 ± 26.5 km, while the mean distance between the FTIR site and the AirCore landing point is 101.1 ± 28.5 km. The same approach is undertaken when comparing with aircraft observations but the aircraft profiles generally extend up to around 6 km asl altitude whereas the aircore profiles extend up to 20 km or higher. As mentioned in Section 2.3, aircraft observations are conducted approximately 100 km northeast of the FTIR site."*